



# The effects of surface roughness on the spectral (300 – 1400 nm) bidirectional reflectance distribution function (BRDF) of bare sea ice

Maxim L. Lamare[1], John D. Hedley[2], and Martin D. King[1]

[1]Department of Earth Sciences, Royal Holloway University of London, Egham, Surrey, TW20 0EX, UK
[2]Numerical Optics Ltd, Witheridge, Tiverton, Devon, EX16 8AA, UK

*Correspondence to:* M.D. King (m.king@rhul.ac.uk)

**Abstract.** The Bidirectional Reflectance Distribution Function (BRDF) has been calculated for three types of bare sea ice with varying surface roughness ($\sigma = 0.1$–10 cm) and ice thicknesses (50–2000 cm) over an incident solar irradiance wavelength range of 300–1400 nm. The comprehensive study of the BRDF of sea ice presented here is paramount for interpreting sea ice measurements from satellite imagery and inter-calibrating spaceborne sensors that derive albedo from multiple multi-angular measurements. The calculations performed by a radiative transfer code (PlanarRad) show that the BRDF of sea ice is sensitive to realistic values of surface roughness. The results presented here show that surface roughness cannot be considered independently of sea ice thickness, solar zenith angle and wavelength. A typical BRDF of sea ice has a quasi-isotropic reflectance over the hemisphere, associated with a strong forward scattering peak of light. Surface roughness is crucial for the location, size and intensity of the forward scattering peak. As the surface roughness increases, a spreading of the BRF peak is observed. The peak remains specular for the smaller surface roughnesses ($\sigma = 0.001$ m to $\sigma = 0.01$ m), whereas for larger surface roughness features (above $\sigma = 0.05$ m), the peak spreads out over multiple quads with a lower intensity than for smaller roughness features, and the highest value is displaced further out on the solar principal plane. Different types of sea ice have a similar pattern with wavelength, the BRF increases by 29.5% from first-year sea ice to multi-year sea ice at 400 nm and up to 630.7% at 1100 nm, 31.5% from melting sea ice to multi-year sea ice at 400 nm and a maximum of 97.7% at 900 nm, and 11.3% from melting sea ice to first-year sea ice at 400 nm and up to 86.2% at 800 nm. As a specific example, for first-year sea ice at $\lambda = 500$ nm and $\theta_i = 60°$, the BRF of an optically thick layer with surface roughness of $\sigma = 0.001$ m is 0.543 at nadir. The forward scattering peak is spread over a single quad located at $\phi_r = 0°$, $\theta_r = 60°$, that has a BRF of 9.748. For the same configuration with surface roughness of $\sigma = 0.1$ m, the nadir has a BRF of 0.549 and the forward scattering peak is spread over 18 quads, located between $\phi_r = 345°$ and $\phi_r = 15°$, $\theta_r = 40°$ and $\theta_r = 87.5°$ with values between 0.776 and 5.089. The BRDF calculations presented in this study form the first set of complete BRDF values for bare sea ice with a wide range of configurations.

## 1 Introduction

Knowledge of the surface albedo of sea ice and its temporal variability is essential to understand the energy budget of polar regions, that strongly affects the Earth's climate system (e.g. Curry et al., 1995; Qu and Hall, 2005; Flanner et al., 2011). Sensors aboard Earth Observing satellites allow the synoptic observation of expansive areas with regular repeat coverage, providing



an ideal tool for the monitoring of albedo at high latitudes(e.g. Bacour et al., 2020; Qu et al., 2015). However, the scattering of light from the surface of sea ice is not isotropic (e.g. Buckley and Trodahl, 1987) and therefore calculations of spectral albedo rely on the knowledge of viewing and illumination angles. Most satellite sensors are only able to measure reflected energy over a small number of viewing angles and spectral bands. Indeed, only a limited number of satellite systems currently

provide near-simultaneous multi-angular measurements (Gatebe and King, 2016) and satellite sensors commonly used to derive surface albedo such as MODIS (Moderate Resolution Imaging Spectroradiometer), are constrained to collecting multi-angular measurements over several orbits. Therefore, knowledge of the angular distribution of the reflected radiation of sea ice is necessary to accurately derive surface albedo and provide climate models rwith reliable inputs. The Bidirectional Reflectance Distribution Function (BRDF) is a directional description of albedo, describing the relationship between illumination and

viewing angles (Nicodemus et al., 1977). Previous studies have characterised BRDF of snow, either with field measurements of the Hemispherical Directional Reflectance Function (HDRF) as an approximation of BRDF (e.g. Middleton and Mungall, 1952; Dirmhirn and Eaton, 1975; Dozier et al., 1988; Bourgeois et al., 2006; Malinka et al., 2016), focussing on roughness(e.g. Manninen, 1997), polarization(e.g Leroux et al., 1998; Peltoniemi et al., 2005a, 2009), physical properties like grain size and sastrugi(e.g Aoki et al., 2000; Stanton et al., 2016; Kuchiki et al., 2011; Sun and Zhao, 2011), wavelength(e.g Painter and

Dozier, 2004; Peltoniemi et al., 2005b; Dumont et al., 2010; Ehrlich et al., 2012), impurities(e.g Peltoniemi et al., 2015), fine angular resolution(e.g. Goyens et al., 2018) or specific places like Antarctica(e.g Hudson et al., 2006; Marks et al., 2015; Carlsen et al., 2020), Arctic(e.g Ball et al., 2015; Hakala et al., 2014), glacial(e.g. Joerg et al., 2015)or specifically snow on sea sea ice(e.g. Heygster et al., 2012a; Li and Zhou, 2004a, b; Lyapustin et al., 2010) There have also been studies modelling BRDF or the Bidirectional Reflection Function (BRF) as an approximation of BRDF (e.g. Leroux et al., 1999; Mishchenko

et al., 1999; Dumont et al., 2010; Heygster et al., 2012b; Kokhanovsky and Breon, 2012b; Xiong and Shi, 2014). The BRDF of glacial ice has been reported by (e.g. Ren et al., 2021) and lake ice by (e.g. Miao et al., 2020). BRDF models of snow are found concentrating on mountain snow (e.g. Dozier and Painter, 2004; Painter et al., 2013; Wang et al., 2017), snow kernals(e.g. Ding et al., 2019b, a; Jiao et al., 2019; Qu et al., 2016b), snow grain size(e.g. Zege et al., 2011), polarization(e.g. Yang et al., 2019; Xiong and Shi, 2014), sastrugi(e.g. Corbett and Su, 2015), remote sensing(e.g Pope et al., 2016; Wu et al., 2012, 2011; Jin and

Simpson, 2001) and models (e.g Kokhanovsky and Breon, 2012a; Hudson et al., 2010). BRDF models also exits for seaice (e.g. Qu et al., 2016a; Mishchenko et al., 1999)

The effect of light absorbing impurities on HDRF (e.g. Khan et al., 2017; Li and Zhou, 2002) and angular reflectivity of snow(e.g. Lv et al., 2018) have also been reported. The BRDF of snow covered sea ice has also been measured or modelled (e.g. Arnold et al., 2002; Li and Zhou, 2004b), but the characterisation of the BRDF of bare sea ice in the literature remains scarce.

Jin and Simpson (1999) calculated the Anisotropy Reflectance Factor (ARF) for bare sea ice. The ARF is equivalent to the ratio of the isotopic albedo to measured albedo and is measure of the similarity (or not) to an isotropic reflected radiation field(Jin and Simpson, 1999). Jin and Simpson (1999) showed that sea ice has a larger reflectance in the forward observation direction and presents a high sensitivity to solar elevation and surface roughness. However the study was limited to 2 spectral bands at 580–680 nm and 725–1000 nm and a single type of multi-year sea ice with parameters obtained from Weeks and Ackley (1994).

Schlosser (1988) measured the angular reflected radiance of laboratory grown sea ice for varying ice thicknesses between 6



mm and 11 cm, showing a strong dependance of BRDF on ice thickness and structure. Arnold et al. (2002) and Gatebe and King (2016) described airborne BRDF measurements acquired for a variety of natural surfaces over 13 bands from 502 nm to 2289 nm, including polar snow and sea ice. The BRDF of snow-covered sea ice, melt-season sea ice and snow-covered tundra were reported for a limited number of solar zenith angles, showing quasi-isotropic reflectance outside an enhanced forward

scattering peak. Stamnes et al. (2011) modelled the BRDF of snow covered and bare sea ice, using a coupled atmosphere-snow-ice-ocean radiative-transfer model. Using sea ice inherent optical properties (IOPs), Stamnes et al. (2011) computed the BRDF for a range of sea ice types between 300 and 4000 nm. The theoretical computations relied on a smooth interface between the media however, and to represent surface roughness, the authors used a fixed 10° Gaussian beam, that did not take in account varying surface roughness effects which have been shown to significantly affect BRDF (Jin and Simpson, 1999).

Owing to its complex nature, the optical and physical properties of sea ice vary spatially and temporally, altering the solar radiation reflected from the surface (Perovich, 1996). Previous studies have demonstrated a strong dependence of albedo on the type of sea ice (e.g. Perovich et al., 2002; Marks et al., 2015). Light reflection and transmission are sensitive to changes in the thickness of the sea ice (Perovich, 1996), and surface roughness has been shown to significantly affect the angular pattern of reflectance at larger viewing angles for snow (Warren et al., 1998; Ball et al., 2015) and sea ice (Jin and Simpson, 1999). Yet,

to the authors' knowledge, no modelling studies comprehensively characterising the BRDF of bare sea ice have been carried out previously. Thus, a systematic study of the dependance of the BRDF of multiple types of bare sea ice to changing surface roughness conditions and varying thickness is required.

In this work, the radiative-transfer model PlanarRad (Hedley, 2008, 2015) was used to model the BRDF of three different types of sea ice from 300 to 1400 nm with varying thicknesses as a function of surface roughness in two steps. Firstly, the

BRDF of three different types of sea ice with a thickness large enough to be optically thick was modelled with an increasing surface roughness. Secondly, the calculations performed in the first step were repeated, but the optically thick thicknesses were replaced with fixed thicknesses of 50 cm and 100 cm for each type of sea ice.

## 2   Methods

### 2.1   Definitions

BRDF is commonly used to represent the reflective properties of a surface by describing the angular distribution of the scattering of light from the surface (Nicodemus et al., 1977). The spectral BRDF describes the relationship between the irradiance incident from a given direction relative to its contribution to the reflected radiance in another direction (Nicodemus et al., 1977), which can be expressed mathematically by:

$$BRDF(\lambda) = fr(\theta_i, \phi_i; \theta_r, \phi_r; \lambda) = \frac{dL_r(\theta_i, \phi_i; \theta_r, \phi_r; \lambda)}{dE_i(\theta_i, \phi_i; \lambda)}, \tag{1}$$

where $\theta$ and $\phi$ are the zenith and azimuth angles respectively in a spherical coordinate system, $\lambda$ is the wavelength of light, L is radiance, E is irradiance, *i* refers to incident directions and *r* to reflected directions (Nicodemus et al., 1977; Schaepman-Strub et al., 2006). The angles used to define BRDF are shown in Figure 1a. BRDF requires the irradiance to be in the form of





a collimated beam and the radiance to be measured with an infinitesimal solid angle. Thus, BRDF cannot be measured directly (Schaepman-Strub et al., 2006). In order to facilitate comparison with the literature and field studies, the BRDF computed in this study was converted to the unitless Bidirectional Reflectance Factor (BRF). BRF is defined by the ratio of the reflected radiant flux, $d\Phi_r$ from a surface area to the reflected radiant flux, $d\Phi_r^{lamb}$ from an idea Lambertian reflector under identical

viewing angles and single direction illumination (Schaepman-Strub et al., 2006). Therefore, BRF is expressed as:

$$BRF(\lambda) = \frac{d\Phi_r(\theta_i, \phi_i; \theta_r, \phi_r; \lambda)}{d\Phi_r^{lamb}(\theta_i, \phi_i; \lambda)} \qquad (2)$$

The BRDF of an ideal Lambertian reflector is $\frac{1}{\pi}$ (Nicodemus et al., 1977; Schaepman-Strub et al., 2006). Hence, the BRDF calculated by PlanarRad may be converted to BRF by multiplying by $\pi$.

## 2.2   Model description

The calculations of the BRDF of sea ice were performed using PlanarRad (Hedley, 2015), a radiative-transfer model that computes the radiance distributions and derived quantities for homogeneous scattering and absorbing media (Hedley, 2008). The model is an open-source implementation of the invariant imbedded numerical integration technique for radiative-transfer, based on the algorithm described by Mobley (1994). PlanarRad has previously been used for reflectance computations in marine environments (Lim et al., 2009; Hedley et al., 2012) and is functionally similar to the commercial software Hydrolight

(Mobley, 1989).

In PlanarRad, radiance is calculated as the average radiance over finite solid angles, defined by a discretisation of the surface of a sphere divided into two hemispheres (Figure 1b). The lower hemisphere corresponds to the upwelling radiance (exiting the surface), whereas the upper hemisphere corresponds to the downwelling direct sky radiance. The discretisation is determined by bounding lines of constant zenith ($\theta$) and azimuth ($\phi$) angle, forming quadrangular regions, commonly called "quads". The

two hemispheres are divided into 9 by 24 segments each, forming a total of 432 quads over the whole sphere. The directionally averaged radiance is computed by PlanarRad within each quad. The input irradiance is set to a single quad with a fixed azimuth, $\phi_i$ and a variable zenith, $\theta_i$, the model being rotationally invariant. For the rough surface constructed from randomly oriented surfaces used in this study, only the relative azimuth angle between $\phi_i$ and $\phi_r$ is required. The incident irradiance was fixed at a constant value for the purpose of this study. The azimuth angles corresponding to the quad centres are located every $15°$ from

$\phi = 0°$ to $\phi = 345°$ and the zenith angles corresponding to the quad centres are located at $\theta = 0, 10, 20, 30, 40, 50, 60, 70, 80$ and $87.5°$. Out of convention, the incident azimuth angle, $\phi_i$ was set to $180°$, the quarter-sphere from 270 to $90°$ azimuth representing the forward scattering of light and the quarter-sphere from 90 to $270°$ azimuth representing the backward scattering. Thus, the solar principal plane is defined as $\phi = 180–0°$. Figure 2 shows a typical 2D polar plot of a PlanarRad output for optically thick (as described in Sect. 2.3) first-year sea ice, with a solar zenith angle, $\theta_i = 60°$ and a roughness parameter of $\sigma = 0.01$ m

(described below).

The absorption coefficient, $a$, attenuation coefficient, $\alpha$, scattering phase function, complex refractive index of sea ice, complex refractive index outside the sea ice, surface roughness and thickness of the sea ice were used in the radiative-transfer calculations. The parameters are presented in Sect. 2.3. The calculations presented here assume that no atmosphere is present and that





the sea ice is floating on an optically thick body of sea water that has a wavelength independent diffuse reflectance of 0.1. A roughness parameter effecting the statistical distribution of surface slope was implemented in a similar way to that described in Mobley (1994). The roughness parameter, $\sigma$, describes the standard deviation of the height relative to the horizontal distance, and is therefore unitless. For example, if $\sigma = 1$, the slope between points located 1 mm apart has a standard deviation of 1 mm. As the system is considered spatially consistent, the overall BRDF and the effect of $\sigma$ is scale invariant. The surface was modelled as a grid of equilateral triangles and the height of the vertices was set randomly using $\sigma$. The procedure is the same as the one applied to water surfaces in Mobley (1994), except there $\sigma$ is derived from windspeed and the triangles are not equilateral to account for directional dependancy of water waves. The light transfer across the realised surfaces was modelled using Monte Carlo ray tracing, over the discretised sphere described previously. In the work presented here, 5 modelled surfaces were generated with an elevation standard deviation, $\sigma = 0.001, 0.005, 0.01, 0.05$ and $0.1$ (Figure 3). The surfaces were generated using 10 rays per quad (4320 rays in total) with results averaged over 2000 surfaces. The roughness model being scale invariant, and the relative amplitude defined as 1 meter, the scale height of the roughness is 0.1, 0.5, 1, 5 and 10 cm.

## 2.3 Calculation of the BRF of 3 types of sea ice with different roughness parameters

The BRF of three types of sea ice were modelled: first-year ice, multi-year ice and melting ice. The selected optical and physical parameters were based on field studies and cover a wide range of observed values. A base amount of black carbon was added to the model to be more representative of natural sea ice, as small quantities of black carbon deposited from the atmosphere in polar regions (e.g Doherty et al., 2010) are likely to be found in sea ice. The mass absorption coefficient of black carbon was calculated using Mie theory, using refractive indices from Chang and Charalampopoulos (1990), and following the method described by Flanner et al. (2007). A mass-ratio of 1 ng g$^{-1}$ of black carbon was added to the sea ice, by combining the mass absorption coefficients of sea ice and black carbon. The attenuation coefficient of sea ice was calculated using the scattering cross-sections and densities described by Lamare et al. (2016) and Marks and King (2014), as:

$$\alpha(\lambda) = a(\lambda) + s(\lambda); \quad s(\lambda) = \phi_s \sigma_s, \tag{3}$$

where $\alpha$ is the attenuation coefficient of sea ice, $a$ is the absorption coefficient of sea ice with an added mass-ratio of 1 ng g$^{-1}$ of black carbon, $s$ is the scattering coefficient of sea ice, $\phi_s$ is the scattering cross-section and $\sigma_s$ is the density. According to Light et al. (2004), the fractional volume of ice is larger than the fractional volume of brine, and the absorption coefficient of ice is similar to the absorption coefficient of brine, hence the absorption coefficient of sea ice may be set equivalent to pure ice. Therefore, the refractive index of pure ice (Warren and Brandt, 2008) was used for sea ice and a refractive index value of 1.0 was used above sea ice. To describe the directionality of the scattering of the sea ice, the Henyey-Greenstein phase function (Henyey and Greenstein, 1941) was used, with a fixed, wavelength independent asymmetry factor $g$ of 0.98 (Lamare et al., 2016). In this work, the asymmetry parameter, $g$, and the attenuation coefficient, $a$ were held constant, and the scattering coefficient, $s$ was varied to simulate different sea ice configurations, according to the methods outlined in Lee-Taylor and Madronich (2002). The optical and physical parameters of the selected sea ice types are summarised in Table 1. The scattering





coefficient was fixed with wavelength (Malinka et al., 2018; Lamare et al., 2016).

The BRDF of the three different types of sea ice were subjected to solar radiation with a wavelength from 300 to 1400 nm with a 100 nm interval, as a function of surface roughness and thickness. The solar zenith angle was varied in 10 steps corresponding to the centre of the quads, from $\theta_i = 0°$ to $\theta_i = 87.5°$, and the surface roughness parameterisations described in section 2.2 were

used, providing a wide range of configurations.

In some of the experiments described here, the sea ice was defined as optically thick, to allow for a direct comparison between the different types of ice and with studies present in the literature. An optically thick sea ice as defined in this study as a sea ice with a thickness for which the underlying medium (i.e. seawater) does not affect the surface reflectance. The sea ice was considered to be optically thick between 3 to 5 $e$-folding depths, *i.e.* where over 95% of the incident light is attenuated (France

et al., 2011). An optically deep thickness of 1.85 m for first-year sea ice, 3.75 m for multi-year sea ice and 20 m for melting sea ice were picked, based on values compiled by Lamare et al. (2016). In a second step, sea ice thicknesses of 50 cm and 100 cm were selected for the three different types of sea ice. The two thicknesses were chosen to examine and inter-compare the effect of the sea ice thickness and roughness on the BRDF of different sea ice types rather than model representative values. Nevertheless, the model can produce results for a range of thicknesses, from the centimetre scale to optically thick thicknesses.

## 3   Results

### 3.1   Nadir BRF of sea ice and forward scattering peak

The nadir BRF of the three types of sea ice was computed with varying thicknesses from 300 nm to 1400 nm and for a fixed solar zenith angle of $\theta_i = 60°$. In a second step, the forward principal plane of optically thick first year sea ice was plotted as a function of solar zenith angle for a range of surface roughness values.

### 3.1.1   Nadir BRF of sea ice for varying sea ice thicknesses and surface roughness

The nadir BRF of first-year sea ice, multi-year sea ice and melting sea ice with thicknesses 50 cm, 100 cm and the optically thick thicknesses are shown in Figure 4. The plotted data were obtained from the nadir quad of PlanarRad, with a surface roughness of $\sigma = 0.01$ m, and a solar zenith angle $\theta_i = 60°$. For the three types of sea ice, the BRF is strongly wavelength dependent due to the large absorption in the ice dominating the signal beyond 700 nm, and significantly lowering the BRF.

Although the different types of sea ice have a similar pattern with wavelength, the BRF increases by 29.5% from first-year sea ice to multi-year sea ice at 400 nm and up to 630.7% at 1100 nm, 31.5% from melting sea ice to multi-year sea ice at 400 nm and a maximum of 97.7% at 900 nm, and 11.3% from melting sea ice to first-year sea ice at 400 nm and up to 86.2% at 800 nm. The effect of the thickness of the sea ice on the BRF varies according to the type of sea ice. The BRF decreases by 20.6% when going from an optically thick first-year sea ice to a 100 cm thick first-year sea ice and 47.4% from optically thick to 50 cm. For

multi-year ice the decrease in BRF is 3% from optically thick to 100 cm and 12.8% from an optically thick thickness to 50 cm. Melting sea ice shows the largest change in BRF relative to thickness with a decrease in BRF of 72.9% between an optically



thick thickness and 100 cm and 83.6% between an optically thick thickness and 50 cm. Melting sea ice is more translucent than first-year or multi-year sea ice, therefore more light penetrates the sea ice deeper and is absorbed by the underlying seawater, explaining the larger reduction in BRF at nadir. On the contrary, with sea ice types that scatter light more efficiently, less light penetrates the ice and the proportion absorbed by the seawater under the ice is less.

### 3.1.2 The effects of roughness on the forward scattering peak of the BRF

To investigate the influence of surface roughness on the location of the dominant directional scattering of light, hereafter referred to as forward scattering peak, the BRF along the solar principal plane is presented. Knowledge of the intensity and size of the forward scattering peak are essential to reliably calculate the energy budget of the sea ice, and correct for the fluctuations in temporal remote sensing data (e.g. Leroy and Roujean, 1994; Li et al., 1996). Figure 6 shows the effects of surface roughness on the forward scattering peak of the BRF of optically thick, first-year, sea ice with a solar zenith angle, $\theta_i = 60°$. The results are also representative of multi-year and melting sea ice. Figure 6a displays the intensity, shape and position of the BRF peak on the forward solar principal plane ($\phi_r = 0°$). As the surface roughness increases, a spreading of the BRF peak is observed. Indeed, the peak remains specular for the smaller surface roughnesses ($\sigma = 0.001$ m to $\sigma = 0.01$ m), whereas for larger surface roughness features (above $\sigma = 0.05$ m), the peak spreads out over multiple quads with a lower intensity than for smaller roughness features, and the highest value is displaced further out on the solar principal plane. Figure 6b shows the effect of surface roughness on the position of the BRF peak on the solar principal plane under different illumination conditions ($\theta_i = 0$ to $87.5°$). For the smaller roughness features ($\sigma = 0.001$ m to $\sigma = 0.01$ m), the position of the peak on the solar principal plane is specular and therefore matches the solar zenith angle. A roughness of $\sigma = 0.05$ m affects the position of the BRF peak at low sun angles ($\theta_i = 60$ to $87.5°$), moving the peak to a lower position on the hemisphere and therefore to a higher viewing zenith angle. For a solar zenith angle $\theta_i = 60°$, the viewing zenith angle is $\theta_r = 70°$, for $\theta_i = 70°$, $\theta_r = 80°$ and for $\theta_i = 80°$ and $87.5°$, $\theta_r = 87.5°$. With a surface roughness of $\sigma = 0.1$ m, the forward scattering peak is located at higher viewing zenith angles than the solar zenith angles, except for $\theta_i = 10$ and $20°$, where the angle of the forward scattering peak equals the angle of incident illumination.

### 3.2 The BRF of sea ice with varying surface roughness

#### 3.2.1 BRF and thickness

Figure 5 shows the BRF of first-year, multi-year and melting sea ice with a solar zenith angle, $\theta_i = 60°$ and at a wavelength, $\lambda = 500$ nm. The BRF was modelled for three thicknesses as a function of surface roughness: 50 cm, 100 cm and an optically thick layer. The modelled BRF pattern is similar to snow (e.g. Dumont et al., 2010) and consistent with the literature for sea ice (e.g Arnold et al., 2002), showing a quasi-isotropic reflectance apart from a strong forward scattering peak. The surface roughness plays an essential role in the BRF of sea ice, by controlling the location and size of the forward scattering peak, as shown previously in Figure 6. Indeed, the peak is mostly specular and located in a single quad for a surface roughness of $\sigma = 0.001$ m and spreads out over multiple quads and moves lower on the hemisphere with a larger surface roughness. As a



specific example, for first-year sea ice at $\lambda = 500$ nm and $\theta_i = 60°$, the BRF of an optically thick layer with surface roughness of $\sigma = 0.001$ m is 0.543 at nadir. The forward scattering peak is spread over a single quad located at $\phi_r = 0°$, $\theta_r = 60°$, that has a BRF of 9.748. For the same configuration with surface roughness of $\sigma = 0.1$ m, the nadir has a BRF of 0.549 and the forward scattering peak is spread over 18 quads, located between $\phi_r = 345°$ and $\phi_r = 15°$, $\theta_r = 40°$ and $\theta_r = 87.5°$ with

values between 0.776 and 5.089. Furthermore, the effects of thickness and surface roughness on the BRF of sea ice are inter-dependant. For smaller surface roughness parameters, an increase in the thickness of the sea ice mainly changes the intensity of the quasi-isotropic part of the BRF, affecting the forward scattering peak much less. For the first-year sea ice with the configuration described above and a roughness parameter of $\sigma = 0.001$ m, the BRF of the quad with the highest value in the specular peak increases by 2.42% from a 50 cm layer to an optically thick layer whereas the BRF at nadir increases by 45.57%.

For larger surface roughnesses, a change in sea ice thickness affects the specular peak strongly, as well as the quasi-isotropic part of the BRF. The BRF of the first-year sea ice described previously with a surface roughness of $\sigma = 0.1$ m changes by 81.66% at nadir and between 3.27% and 69.07% in the forward scattering peak between a layer of 50 cm and an optically thick layer. Thus, the distribution and values of the BRF over the azimuth, $\phi$ and zenith, $\theta$ are sensitive to the thickness and the roughness.

### 3.2.2   BRF and solar zenith angle

Figure 7 shows the BRF of optically thick first-year, multi-year and melting sea ice at $\lambda = 500$ nm, with an increasing surface roughness for three solar zenith angles, $\theta_i = 50°$, $\theta_i = 70°$ and $\theta_i = 80°$. The results for $\theta_i = 60°$ can be found in Figure 5 for comparison. Low illumination angles (large solar zenith angles) are presented in this study, as they are representative of conditions observable in polar regions. The location and intensity of the forward scattering peak are strongly influenced by

the incident zenith angle, whose effects are inter-dependant of surface roughness. For a small surface roughness of $\sigma = 0.001$ m, the highest value of the forward scattering peak is equal to the incident illumination angle over the range of solar zenith angles, however the intensity of peak increases with $\theta_i$. For first-year sea ice, the peak BRF increases from 5.01 for $\theta_i = 50°$ to 28.92 for $\theta_i = 70°$ and to 143.16 for $\theta_i = 80°$. The forward scattering peak diffuses with larger solar zenith angles, from 1 quad at $\theta_i = 50°$ to 3 quads at $\theta_i = 80°$ for all three types of sea ice. With surface roughnesses of $\sigma = 0.005$ m and $\sigma = 0.01$

m, the forward scattering peak increases in intensity with increasing solar zenith angles, however the peak remains spread over a similar number of quads between $\theta_i = 50°$ and $\theta_i = 80°$. For larger surface roughnesses of $\sigma = 0.05$ m and $\sigma = 0.1$ m, although the intensity of the wide forward scattering peak increases with larger solar zenith angles, the intensity is lower than for small roughness parameters. For first-year sea ice with a surface roughness of $\sigma = 0.1$ m, the highest BRF value is 1.360 for $\theta_i = 50°$ and 54.271 for $\theta_i = 80°$. Moreover, the forward scattering peak is distributed over a larger number of quads for

higher incident illumination angles. At large solar zenith angles, typical of polar latitudes, the isotropic part of the BRF remains similar with an increasing surface roughness, whilst the forward scattering peak diffuses and moves to larger viewing zenith angles than the incident illumination angles.



### 3.2.3 BRF and wavelength

The BRF of optically thick first-year sea ice, multi-year sea ice and melting sea ice with increasing surface roughness, for a solar zenith angle $\theta_i = 60°$ and for wavelengths of $\lambda = 400$ nm, 800 nm and 1300 nm is shown in Figure 8. The results for $\lambda = 500$ nm can be found in Figure 5 for direct comparison. As partly shown in Figure 4, the BRF of sea ice is strongly wavelength dependent. At nadir, the highest BRF values are found in the near ultra violet and visible wavelengths, decreasing rapidly

between 500 and 900 nm. Beyond 900 nm for first-year and melting sea ice and 1000 nm for multi-year sea ice, the BRF tends to zero, owing to the absorption by the sea ice. However, the BRF does not decrease uniformly over the hemisphere with an increasing wavelength. The quasi-isotropic part of the hemisphere follows the same trend as the nadir, whereas the forward scattering peak conserves high BRF values, independently of the wavelength. The behaviour is valid for the entire range of roughness parameters. For optically thick first-year sea ice with a solar zenith angle, $\theta_i = 60°$, the nadir BRF decreases by

99.92% from 400 nm to 1300 nm for a surface roughness of $\sigma = 0.001$ m, and by 99.90% for a surface roughness of $\sigma = 0.1$ m. However, the change within the forward scattering peak with wavelength differs for different amounts of surface roughness. The forward scattering peak located at $\phi_r = 0°, \theta_r = 60°$ for first-year sea ice with a surface roughness of $\sigma = 0.001$ m, decreases by 13.94%. For the same configuration with a surface roughness of $\sigma = 0.1$ m, the wider forward scattering peak decreases non-uniformly and reduces in size. Within the 18 quads of the forward scattering peak located between $\phi_r = 345°$

and $\phi_r = 15°, \theta_r = 40°$ and $\theta_r = 87.5°$, the highest BRF value ($\phi_r = 0°, \theta_r = 87.5°$) decreases by 14.92%, but the lowest BRF value ($\phi_r = 15°, \theta_r = 40°$) decreases by 83.05% between 400 and 1300 nm. The same behaviour is observable for multi-year and melting sea ice. For small roughnesses ($\sigma \leqslant 0.01$ m) the intensity of the forward scattering peak that does not change in size varies little with wavelength compared to the quasi-isotropic part of the BRF. For large roughnesses ($\sigma > 0.01$ m) the forward scattering peak decreases strongly around the edges with wavelength, whereas the centre quads vary by a small amount

as with smaller roughnesses. Furthermore, the quasi-isotropic part of the BRF behaves in the same manner than for smaller surface roughnesses.

## 4   Discussion

### 4.1   The effects of surface roughness on the BRF of sea ice

As shown in Sect. 3, surface roughness plays a paramount role in the BRF of bare sea ice. Not only does surface roughness

have an effect on the reflected radiance, particularly in the forward scattering peak, it also modifies the behaviour of the BRF with other controlling parameters such as thickness, solar zenith angle or wavelength. Surface roughness alone principally changes the specular forward scattering peak by diffusing it around the specular point and outwards to a larger zenith angle. Indeed, a smooth surface reflects the incident light specularly, whereas reflection from a roughened surface is composed of the specular reflection of the angled facets in multiple directions as well as a diffuse component from the multiple reflections

among the facets (Torrance and Sparrow, 1967). A reduction in thickness of a sea ice layer with a small amount of surface roughness mainly decreases the BRF in the quasi-isotropic part, having little effect on the specular peak (Figure 5). Indeed, with



a thinner sea ice layer, a number of the scattered photons are absorbed by the strongly absorbing underlying layer (reflectance of 0.1) instead of exiting the medium upwards. Most of the light scattered forwards exits the sea ice in the same manner as for an optically thick layer, explaining the smaller reduction in BRF for the forward scattering peak. When surface roughness is included, the forward scattering peak is more sensitive to a changing thickness. With an increasing solar zenith angle, the BRF with a smaller roughness parameter shows a decrease in intensity of the BRF over the whole hemisphere apart from the

specular peak that increases and moves in a specular manner relative to the solar zenith angle. With a higher solar zenith angle (lower sun on the horizon), the photons travel less deep into the sea ice than for a lower solar zenith angle and go through fewer scattering events due to the shorter path length and the relative angle between the incident light path and the surface. Therefore, the light is less scattered in multiple directions (lower BRF over the hemisphere) and more light is scattered forwards (stronger specular peak). However increasing the surface roughness introduces more scattering events, as the light is reflected

at different angles off the features. Less photons travel directly in a specular manner, reducing the increase in the forward scattering peak with an increasing solar zenith angle, and the larger number of scattering events lead to a smaller reduction in the BRF of the remaining hemisphere. Miao et al. (2020) also noted forward scattering of light was strongly affected by "ice surface condition". The BRF of sea ice is strongly wavelength dependent owing to the light scattering and absorption by the ice. At shorter wavelengths (300 – 900 nm), sea ice is highly scattering, whereas from 900 – 1400 nm the absorption by the

ice dominates, with a nadir BRF close to zero (Figure 4). For a small amount of surface roughness, the BRF exhibits the same wavelength dependence over the hemisphere, bar the specular peak (Figure 8). Indeed, at longer wavelengths, the photons that are scattered in the sea ice are more likely to be absorbed than the photons quickly exiting the medium in a specular direction, creating a strong anisotropy. With increased surface roughness, a similar trend to the smaller surface roughness is observable, however the size of the forward scattering peak decreases with wavelength. The reduction may be caused by the absorption of

photons at larger wavelengths that would otherwise have exited the ice in a forward direction after a low number of scattering events within the roughness features at lower wavelengths.

In the literature, a similar behaviour of the response of BRF to an increasing roughness was observed by Jin and Simpson (1999), when modelling the anisotropic reflectance factor of sea ice with three different roughnesses. Jin and Simpson (1999) also modelled the effects of a varying solar zenith angle on the anisotropic reflectance factor of sea ice for a fixed roughness,

showing that the reflectance anisotropy is much larger for a solar zenith angle of 60° than for 45°. The effect of the solar zenith angle on the angular reflectance of sea ice was confirmed in the work presented here, additionally showing that surface roughness modulates the intensity and width of the forward scattering peak. Arnold et al. (2002) presented a airborne case measurement of BRF for melt-season ice with a solar zenith angle $\theta_i = 55°$. Their measurement of BRF shows no significant departure from uniformity across the hemisphere, apart from a forward scattering peak spread widely forward of the specular

peak, suggesting large surface roughness. The BRF pattern, as well as the BRF reported in the principal plane for melt-season sea ice are in agreement with the modelled BRF presented here. However, the results from Arnold et al. (2002) are not directly comparable with the modelled BRF, as the irradiance for the BRF measured with the Cloud Absorption Radiometer instrument is composed of a direct and a diffuse component, whereas the illumination in the modelling conducted here is direct only.



Although not bare sea ice, Manninen et al. (2021) noted BRF of snow and found surface roughness increased back scattering at large solar zenith angles and Carlsen et al. (2020) found MODIS MCD43 underestimate anisotropy of surface reflection.

## 4.2 Defining the roughness parameter

Sea ice roughness shows significant spatial variability, with vertical features ranging from the millimetre-scale to the meter-scale (e.g. Manninen, 1997; Peterson et al., 2008). The larger surface roughness features are generally caused by the deforma-
tion of the sea ice, forming rubble fields and pressure ridges that can reach 10 to 20 m in height (Tucker et al., 2013). At a smaller scale, brash ice, ridged blocks or frost flowers can create roughness with a standard deviation of a few millimetres to centimetres. As shown in the results (Sect. 3), surface roughness strongly influences the BRDF of sea ice. To cover a wide range of conditions, a selection of five surface roughness parameters, defined by the standard deviation of the height of the surface were picked, with a standard deviation of 1 mm to 10 cm. The range of surface roughness is in agreement with observations
reported in the literature for small scale roughnesses (e.g. Tucker et al., 2013).

Random surface realisations were generated to calculate the surface roughness in the model, which is rotationally invariant. Therefore, Planarrad produces a random surface roughness, that has no specific structure or pattern. Specific complicated shapes present in sea ice, such as pressure ridges were not modelled.

## 4.3 Model limitations

As described in Sect. 2.2, Planarrad computes the BRDF over a hemisphere discretised into quads, and the calculated radiance leaving the surface is averaged over each quad. The input irradiance was set to a single quad in this study. Therefore the angular resolution of the model is limited to the quad size. Any differences in radiance within a single quad cannot be resolved, which results in a loss of definition for features smaller than the size of a single quad. Furthermore, in this configuration, the radiance for a quad containing the forward scattering maxima is lower than the radiance of a specular peak if it is smaller than
a quad. Ideally the solid angle of the illumination source, as well as the solid angle of the quads should tend to zero. However, increasing the discretisation necessitates a considerable computational effort, which led the authors to the current choice of angular resolution representing a balance between resolution and computational resources.

The radiative-transfer equation was computed without an atmosphere, providing a surface BRDF product, whereas the radiance measured by satellite sensors at the Top-Of-Atmosphere (TOA) is a function of the properties of the surface and
the atmospheric conditions at the time of the measurement. The purpose of the study was to characterise and quantify the intrinsic BRDF of sea ice as a function of roughness and thickness that can be incorporated in radiative transfer models by the community. Therefore, to obtain a direct comparison with remote sensing products that have not been corrected for atmospheric effects, the results of this study have to be propagated to the TOA using an additional radiative-transfer model (e.g. Kotchenova et al., 2008).

In this study, the sea ice was modelled as a single homogeneous slab with defined optical properties. The model does not presently allow for the study of multiple layers with different optical properties. However, PlanarRad allows the input of a BRDF as a lower boundary condition, therefore calculations for a layer of snow on the sea ice are possible. The work presented





here focusses on a comprehensive characterisation of the BRDF of bare sea ice which is lacking in the literature, and adding a layer of snow on the sea ice would have added too much complexity. Thus snow was not considered in this study.

For the BRDF calculations described here, black carbon was assumed to be the only external absorber present in the ice. As described in Sect. 2.3, a base mass-ratio of 1 ng g$^{-1}$ was added to the modelled sea ice. Although organic debris, algae, soot, HULIS or mineral dust have an effect of the radiative forcing of sea ice, other light-absorbing impurities other than black carbon were not examined in this study. Additionally, further investigation related to the effects of a varying mass-ratio of light absorbing impurities on the BRF of sea ice is required.

## 5   Conclusions

This study provides a large dataset parameterising the BRDF of bare sea ice, accounting for varying surface roughnesses. The BRDF / BRF of three different types of sea ice was modelled, for a wavelength range of 300 – 1400 nm. The effects of surface roughness were investigated as a function of thickness, solar zenith angle and wavelength. Radiative-transfer calculations show that surface roughness has a significant effect on the BRDF of sea ice, controlling the anisotropy through the forward scattering peak. Furthermore, the surface roughness is inter-dependent of other parameters that determine the BRDF pattern of sea ice, such as thickness, solar zenith angle and wavelength. As predicted by the model, the BRDF of sea ice exhibits a strong forward scattering peak surrounded by a quasi-isotropic response. For small amounts of surface roughness, a reduction in sea ice thickness decreases the quasi-isotropic part of the BRDF, affecting the forward scattering peak very little, the forward scattering peak changes consistently in a specular manner with a varying solar zenith angle while the intensity of the peak increases, and the forward scattering peak is much less wavelength dependant than the surrounding quasi-isotropic part of the hemisphere. For larger amounts of surface roughness, a decrease in thickness affects strongly the entire BRDF, including the forward scattering peak, the intensity of the forward scattering peak increases and moves to larger viewing zenith angles than the solar zenith angles as the latter increase but remains overall lower than for smaller amounts of surface roughness, and the size of forward scattering peak is strongly wavelength dependent. Because surface roughness is inter-dependent of other physical parameters, it is essential to account for roughness in the theoretical calculations of the radiation budget of sea ice.

The study provides a wide range of BRDF for sea ice that cover a diversity of conditions encountered in nature. The data generated here is expected to facilitate the development of more accurate radiative-transfer models used to derive albedo products by the remote sensing community, as well as reduce uncertainties in global climate models.

*Author contributions.* M. L. Lamare performed the modelling, including the data analysis and result preparation, and authored the paper. J. Hedley designed the model, wrote the roughness code, and edited the paper. M. D. King conceived and oversaw the study, and edited the paper.

*Competing interests.* The authors declare that they have no conflict of interest



30 *Acknowledgements.* The output data from this modelling study are stored on the open repository Zenodo, run by the CERN data centre, under a Creative Commons license. The data can be found at DOI 10.5281/zenodo.5733402. MDK and MLL thank NERC for support under Grant NE/K000770X/1



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



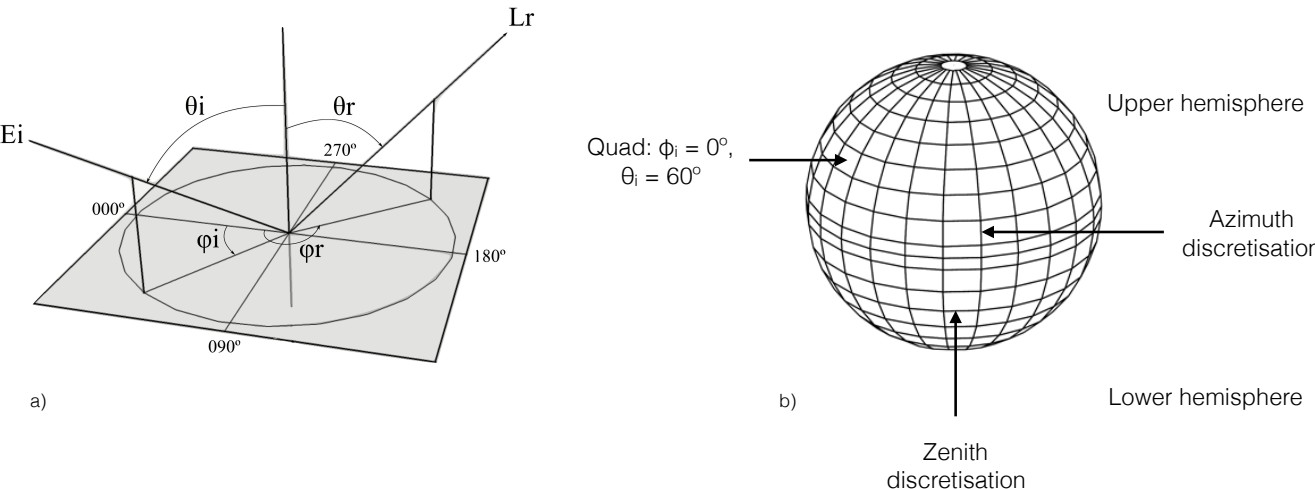

**Figure 1. a)** Diagram of the incident and viewing configuration defining BRDF. $E_i$ is the irradiance from the azimuth angle $\phi_i$ and the zenith angle $\theta_i$. $L_r$ is the radiance in the azimuth angle $\phi_r$ and zenith angle $\theta r$. In this study, $\phi_i$ was fixed to $180°$, the model being rotationally invariant. **b)** Diagram of the directional surface discretisation scheme used by PlanarRad to compute BRDF. Adapted from Hedley (2008).

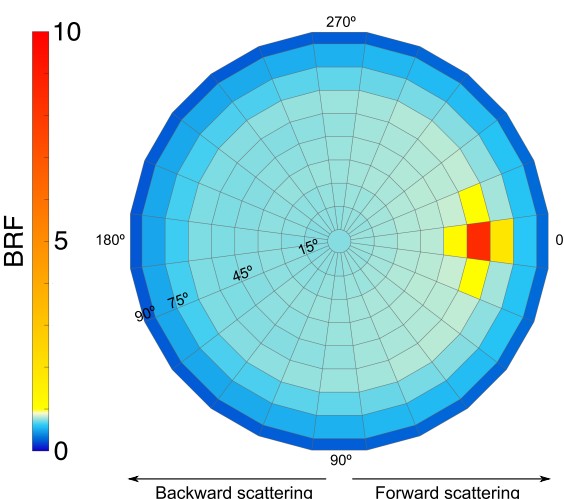

**Figure 2.** Polar plot of the BRF of optically thick first-year sea ice, with a solar zenith angle, $\theta = 60°$ and a roughness parameter of $\sigma = 0.01$ m. The solar azimuth angle $\phi_i$ is located at $180°$, consequently the left half of the hemisphere between $\phi = 90°$ and $\phi = 270°$ represents the backward scattering component and the right half of the hemisphere between $\phi = 270°$ and $\phi = 90°$ represents the forward scattering component. In this case, a strong specular forward scattering peak can be observed centred over the quad located at $\phi_r = 0°$ and $\theta_r = 60°$. A nonlinear colour bar was used to capture the large values around the scattering peak whilst showing the pattern in the quasi-isotropic part of the BRF.





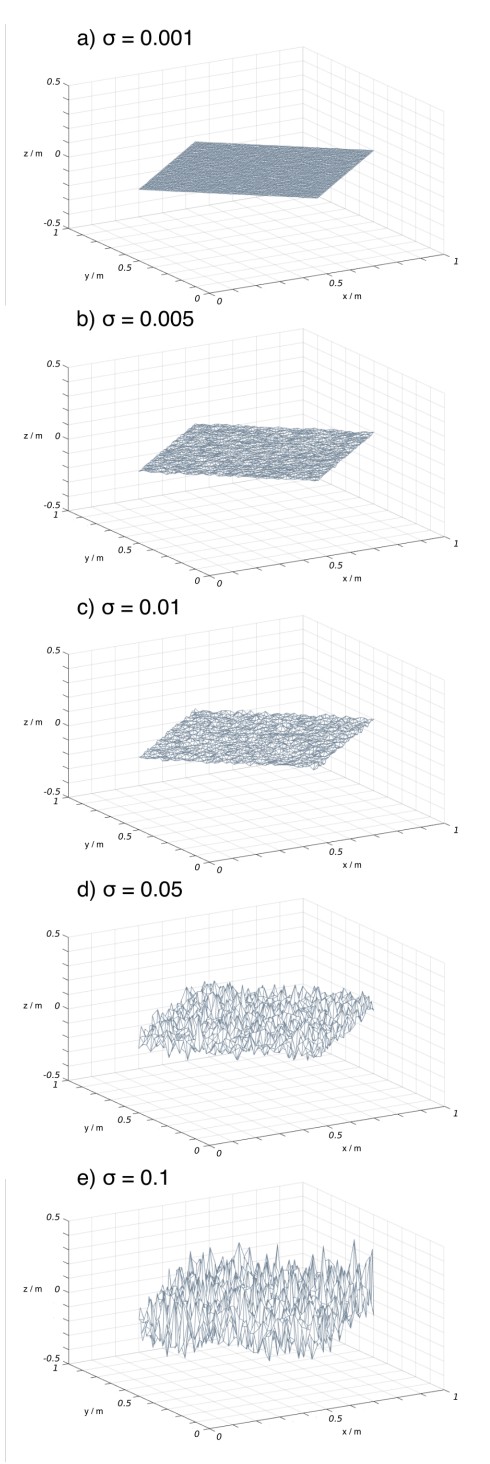

**Figure 3.** Visualisation of example random surface roughness input parameters, controlled by the standard deviation ($\sigma$) of the elevation of the surface. In this study, 5 surface roughnesses of **a)** $\sigma = 0.001$, **b)** $\sigma = 0.005$, **c)** $\sigma = 0.01$, **d)** $\sigma = 0.05$ and **e)** $\sigma = 0.1$ meters were generated.





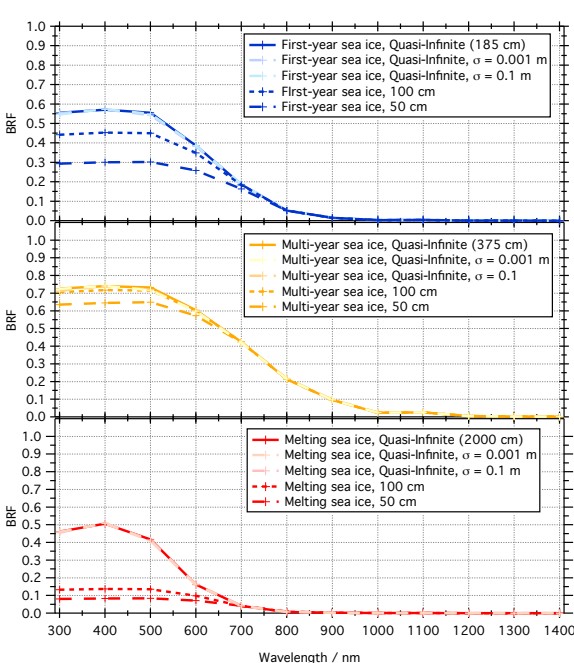

**Figure 4.** Nadir BRF for first-year, multi-year and melting sea ice, from 300 to 1400 nm in 100 nm steps with a solar zenith angle, $\theta = 60°$ and a roughness parameter of $\sigma = 0.01$ m. The BRF of the different types of sea ice is plotted for an optically thick layer (185, 375 and 2000 cm), 100 cm and 50 cm. For each optically thick layer of sea ice, the nadir BRF is plotted for surface roughnesses of $\sigma = 0.001$ m and of $\sigma = 0.1$ m. The different sea ice parameters defined in this study are reported in Table 1



**Figure 5.** BRF of 50 cm, 100 cm and optically thick first-year sea ice, multi-year sea ice and melting sea ice with an increasing surface roughness.The incident angle is $\theta_i = 60°$, and the results are reported for $\lambda = 500$ nm.



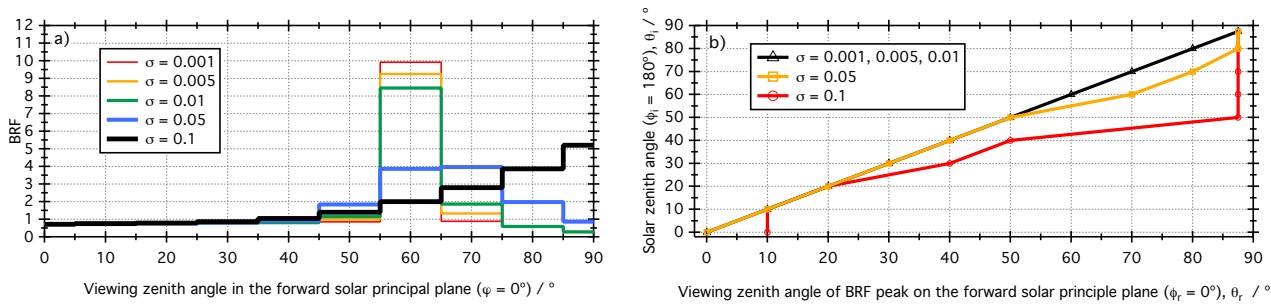

**Figure 6.** The effects of roughness on the forward scattering peak of the BRF. **(a)** BRF in the forward solar principle plane ($\phi_r = 0°$) of optically thick first-year sea ice, modelled with a solar zenith angle $\theta_i = 60°$ as a function of surface roughness. **(b)** Location of the BRF peak of optically thick first-year sea ice on the forward solar principle plane ($\phi_r = 0°$) as a function of solar zenith angle, $\theta_i$ for different surface roughness parameters.





**Figure 7.** BRF of optically thick first-year sea ice, multi-year sea ice and melting sea ice with an increasing surface roughness at $\lambda = 500$ nm. The incident angles are $\theta_i = 50°$, $\theta_i = 70°$ and $\theta_i = 80°$. Note that the scale of the colour bar varies for the different illumination angles in order to visualise clearly the BRF pattern.





**Figure 8.** BRF of optically thick first-year sea ice, multi-year sea ice and melting sea ice with an increasing surface roughness for $\lambda = 400$, $\lambda = 800$ and $\lambda = 1300$. The incident angle is $\theta_i = 60°$.



**Table 1.** Sea ice parameters used as input parameters for the PlanarRad model, based on literature values and detailed in the work of Lamare et al. (2016).

| Sea ice type | Sea ice density (kg m$^{-3}$) | Sea ice scattering coefficient (m$^{-1}$) | Sea ice asymmetry parameter $g$ | optically thick thickness (cm) | Thickness modelled (cm) |
|---|---|---|---|---|---|
| First year sea ice | 800 | 120 | 0.98 | 185 | 50, 100, 185 |
| Multi-year sea ice | 800 | 600 | 0.98 | 375 | 50, 100, 375 |
| Melting sea ice | 800 | 24 | 0.98 | 2000 | 50, 100, 2000 |