# Peer review of "The effects of surface roughness on the calculated, spectral (300 – 1400 nm), conical-conical reflectance factor (CCRF) as an alternative to bidirectional reflectance distribution function (BRDF) of bare sea ice"

_The Cryosphere, 2021_

## Author Comment (AC1)

Reply to Comment on tc-2021-366

Anonymous Referee #1
Referee comment on "The effects of surface roughness on the spectral (300–1400 nm) bidirectional reflectance distribution function (BRDF) of bare sea ice" by Maxim L. Lamare et al., The Cryosphere Discuss., https://doi.org/10.5194/tc-2021-366-RC1, 2022
Within the manuscript, the influence of the surface roughness on the reflectance anisotropy of bare sea ice is simulated for different wavelengths, illumination conditions, and sea ice thicknesses. For this purpose, the authors employ the radiative transfer model PlanarRad. From this, the size and shape of the forward scattering maximum is discussed for different types of sea ice (first-year, multi-year, melting sea ice).

This study marks an important contribution for the remote sensing community, as different retrieval products from aircraft and satellites rely on an accurate knowledge of the multiangular reflectance of sea ice, which so far has been underrepresented in the literature. The retrieval products in need for a better representation of the surface reflectance anisotropy most notably include surface energy budget observations, but also include atmospheric retrievals that rely on surface reflectance corrections (e.g. cloud retrievals).

The manuscript is concise and the figures are generally of a good quality. However, there are some aspects that need further attention in my opinion. After some general comments, the more specific comments and suggestions for technical corrections follow below.

>>>>>>>>>>>>>>>>>>>>>>>>>>>>>>>>>>>>
Thank you.
<<<<<<<<<<<<<<<<<<<<<<<<<<<<<<<<<<<<

General Comments
(1) Nomenclature of reflectance quantities: The manuscript is missing a rigorous and consistent usage of reflectance terms. Even though BRDF and BRF are defined in Section 2.1, the usage throughout the text is inconsistent and, in some cases, erroneous. For example, the title indicates a study focused on the BRDF. Instead, due to the limited angular resolution in the viewing geometry, directional-conical results are presented (compare Schaepman-Strub et al., 2006). Other instances include, e.g. P2L19 (BRF stands for Bidirectional reflectance factor, and BRF is not really an approximation of BRDF, it relates the reflected radiant flux of a sample surface to the radiant flux of an ideal Lambertian surface irradiated under the same conditions, as you define yourself in Sect. 2.1), P2L28 and P3L2 (the BRDF can never be measured), and P2L31(what is an isotropic albedo?). I understand that using BRDF is somewhat established in the literature, however I think the authors need to be more careful and follow the recommendations put forward by Schaepman-Strub et al. (2006) to make an effort to improve the usage of reflectance terms in the literature. This is not limited to the above-mentioned cases, and I suggest the authors check again their usage of terms throughout the manuscript (including the introduction when discussing other studies). You also need to mention that you are not simulating a BRF, but an approximation. If you call it a BRF you are assuming the reflected radiance is constant throughout the viewing quads.

>>>>>>>>>>>>>>>>>>>>>>>>>>>>>>>>>>>>
We agree with the reviewer that paper could be more precise in the definition of the properties evaluated, and we intend to add an initial clarification of this and modify the manuscript for consistency. The model used is designed in the

same way as the model HydroLight which is well-known in the field of hydrological optics. These models work with "quad-averaged radiances" where it assumed the radiance is constant over the solid angle subtended by each segment of the angular discretisation, both for "input" and "output". Therefore the BRDFs evaluated are biconical (geometry equivalent to case 5 in Table 2 of Scheapman-Strub et al. 2006), we will indicate this in the manuscript. Clearly what is modelled (and measured) is an "approximation to the BRDF", which as the reviewer indicates cannot actually be measured, being conceptual. This will be clarified in the manuscripts. The BRF as modelled in the paper is most simply interpreted as the biconical BRDF multiplied by pi, and is also a approximation to the BRF. This will be clarified.

P2L19 - "Bidirectional Reflection Function (BRF) as an approximation of BRDF" will be reworded as "Bidirectional Reflection Function (BRF) as an alternative to the BRDF".

References to "measuring the BRDF" will be changed of "measuring an approximation to the BRDF" or similar.
<<<<<<<<<<<<<<<<<<<<<<<<<<<<<<<<<<<

(2) Introduction: In my opinion, the authors need to elaborate more on the references used in the Introduction. Long lists of references are given for a specific point of discussion, but simply mentioning the reference is not enough. For example, not just mention wavelength (....), but actually mention what the current state of the art is regarding wavelength-dependence of the BRDF/HDRF. So far, the introduction only mentions that different effects on the reflectance anisotropy have been measured before, but these effects are not described at all yet. For a proper state-of-the-art overview, more details need to be given already here. This will also help later to put the results of this study in perspective.

>>>>>>>>>>>>>>>>>>>>>>>>>>>>>>>>>>>
Fixed: We will elaborate more on the references given in the introduction. We had aimed to demonstrate what was known and what was not know briefly.
<<<<<<<<<<<<<<<<<<<<<<<<<<<<<<<<<<<

(3) Results: The structure of the results section needs adjustments, as currently the storyline is hard to follow as effects of roughness, thickness, wavelength and solar zenith angle are mixed throughout the different subsections. In addition, the separation of the Nadir BRF results complicates things in my eyes, as this could be discussed together with the 2D BRF. My suggestion would be: 3.1 Roughness and sea ice thickness (showing Figures 4-6, but maybe even switching the order of Figures 5 and 4), 3.2 Roughness and solar zenith angle (Fig. 7), 3.3 Roughness and wavelength (Fig. 8). That would make the structure easier to follow.

>>>>>>>>>>>>>>>>>>>>>>>>>>>>>>>>>>>
Fixed: The structure of the results section will be adjusted as outlined by the reviewer to make the paper easier to follow.
<<<<<<<<<<<<<<<<<<<<<<<<<<<<<<<<<<<

(4) Related to the last comment, Section 3.2.1 starts describing Figure 5 again as if it is mentioned for the first time in the text. I recommend to restructure and discuss the effect of the thickness already together with the effect of the roughness. First half of 3.2.1 actually discusses roughness again from earlier as well. The influence of the sea ice thickness shows the influence of the underlying surface, i.e. the ocean BRF. Thus, the values are lower as you clearly demonstrate. However, I think it is worth noting in the text that the shape of the BRF itself remains unaltered, meaning the shape and size of the forward scattering peak seems similar for all sea ice thicknesses independent of the sea ice type.

>>>>>>>>>>>>>>>>>>>>>>>>>>>>>>>>>>>>>
Fixed: We will restructure the results section and highlight the apparent lack of change in the forward scattering peak as sea-ice thickness changes
<<<<<<<<<<<<<<<<<<<<<<<<<<<<<<<<<<<<

(5) The authors use the word 'quasi-infinite' in the figures, but 'optically thick' throughout the text. If the more commonly used term 'semi-infinite' would be used consistently, it would make the text much easier to read, and also put the text more in line with the Lamare et al. (2016) terminology. If the authors had any specific reason to name it differently, this should be emphasized more.

>>>>>>>>>>>>>>>>>>>>>>>>>>>>>>>>>>>>>
Fixed: The authors have moved away from the term 'semi-infinite' as it is jargon and non-sensical. We have corrected the figures to report 'optically thick' as opposed to 'semi-infinite' to be more in keeping with the text.
<<<<<<<<<<<<<<<<<<<<<<<<<<<<<<<<<<<<

(6) I agree with the authors that studying the semi-infinite (optically thick) sea ice is important to understand the intrinsic surface BRDF of bare sea ice. However, I feel the study could benefit from including a figure looking at the spectral and solar zenith angle dependence of the BRF for another sea ice thickness that is closer to natural sea ice (as melting sea ice with a thickness of 20 m is more a theoretical consideration). I believe seeing the effects of wavelength and illumination angle on both the theoretical (semi- infinite) and the more realistic (e.g. 50 cm/100 cm) thicknesses would be of interest to many readers. Adding an additional figure could work (maybe restricting to only one roughness value at the other thicknesses), or maybe also adding another column for an additional sea ice thickness in Figs. 7 and 8 could be an option.

>>>>>>>>>>>>>>>>>>>>>>>>>>>>>>>>>>>>>
The data presented in figure 5 for BRF with different thicknesses of sea ice other than optically thick/quasi-infinite thickness would meet this request in part. The paper already has a lot of large figures, and it would take over 100s figures to present all the possible degrees of freedom (solar zenith angle, wavelength, surface roughness and thickness) that could be requested of the dataset. All the data is freely available at doi:10.5281/zenodo.5733402 and any reader can download these data and plot the variations they wish to plot. We wish to add the entire focus of the paper is to explore the effect of surface roughness on the reflectivity of sea ice, not the effect of other variables at constant surface roughness.
<<<<<<<<<<<<<<<<<<<<<<<<<<<<<<<<<<<<

(7) Section 4.2: This is a very important section that puts the choice of the modeled roughness parameter in perspective. However, this should be included in the Methods section already, when mentioning the range of modeled roughness parameters at the end of Sect. 2.2. I also suggest the authors consider making the roughness considerations a separate subsection within the methods, e.g. after 2.2 Model description. As this is a vital part of the study, it should have a separate section in the Methods, and the choice of roughness parameters needs to be motivated already at this point. At the moment, 4.2 seems a bit out of place in the discussion.

\>>>>>>>>>>>>>>>>>>>>>>>>>>>>>>>>>>>

Fixed: We will add a roughness section to the methods and further the description of roughness. This is a good idea.

<<<<<<<<<<<<<<<<<<<<<<<<<<<<<<<<<<<

(8) Please be consistent about the usage of the roughness parameter sigma throughout the manuscript. You first state it is unitless, but then give 'm' as a unit on several occasions, e.g. captions of Figs. 3 and 4, or on P6L23.

\>>>>>>>>>>>>>>>>>>>>>>>>>>>>>>>>>>>

Fixed: The "units" of roughness will be fixed throughout the paper. Our metric for roughness does not have units and will be explained in detail. See reply to reviewer 2.

<<<<<<<<<<<<<<<<<<<<<<<<<<<<<<<<<<<

(9) When you mention specific values of the BRF or changes in %, you give values with 2 or even 3 decimal places, which is an accuracy not needed (with respect to the mentioned model limitations) and makes reading the numbers quite cumbersome. In my view, rounding the values to integer numbers seems to be more than sufficient.

\>>>>>>>>>>>>>>>>>>>>>>>>>>>>>>>>>>>

Fixed: The precision will be smaller and consistent throughout the paper.

<<<<<<<<<<<<<<<<<<<<<<<<<<<<<<<<<<<

Specific Comments
Title: In addition to using the correct reflectance term in the title (see general comments above), I suggest to include modeled/simulated in the title as well so that the reader immediately knows what to expect from the study.

The title will now read

\>>>>>>>>>>>>>>>>>>>>>>>>>>>>>>>>>>>

The effect of surface roughness on the calculated spectral (300-1400nm) biconical reflectance of bare sea ice as an approximation to the bidirectional distribution function (BRF) of bare sea ice

<<<<<<<<<<<<<<<<<<<<<<<<<<<<<<<<<<<

Abstract: the mentioning of quads is not clear as this is not a commonly known term. You either need to define it or express the point you want to make differently. In addition, the angular resolution should be mentioned directly.

>>>>>>>>>>>>>>>>>>>>>>>>>>>>>>>>>>>
Fixed: The word quad has been removed from the abstract to improve readability.
<<<<<<<<<<<<<<<<<<<<<<<<<<<<<<<<<<<

P2L8: please be more accurate with the definition of BRDF at this first instance, it does not describe the relation between illumination and viewing angles, but of the incident and reflected radiation of all sets of illumination and viewing angles.

>>>>>>>>>>>>>>>>>>>>>>>>>>>>>>>>>>>
Fixed: This change will be made. Please see our first reply. We will also include a fuller description of model geometry.
<<<<<<<<<<<<<<<<<<<<<<<<<<<<<<<<<<<

P2L10: add Schaepman-Strub et al. (2006) reference already at this point

>>>>>>>>>>>>>>>>>>>>>>>>>>>>>>>>>>>
Fixed: This change will be made.
<<<<<<<<<<<<<<<<<<<<<<<<<<<<<<<<<<<

P3L14: please avoid statements of novelty in that way

>>>>>>>>>>>>>>>>>>>>>>>>>>>>>>>>>>>
Fixed: The tone will be changed.
<<<<<<<<<<<<<<<<<<<<<<<<<<<<<<<<<<<

P6L17: add 'between ... nm wavelength', as otherwise it could sound like these are the chosen ice thickness values

>>>>>>>>>>>>>>>>>>>>>>>>>>>>>>>>>>>
Fixed: This change will be made.
<<<<<<<<<<<<<<<<<<<<<<<<<<<<<<<<<<<

Fig. 4: the different lines are hard to distinguish, especially for the different roughness values for the quasi-infinite cases

>>>>>>>>>>>>>>>>>>>>>>>>>>>>>>>>>>>
Excellent – this is what Figure 4 is demonstrating. A note will be added in the figure caption to this effect.
<<<<<<<<<<<<<<<<<<<<<<<<<<<<<<<<<<<

Sect. 3.1.1, strongest relative change in nadir BRF for melting sea ice. As far as I understand it, that is to be expected due to the much larger semi-infinite sea ice thickness of about 20 m. I think the authors should elaborate a bit on this and include more details on how these

quantities were calculated in Lamare et al. (2016). P6L8 indicates that you calculated the e-folding depth times 3 or 5. If I compare to Table 1 in Lamare et al. (2016), it seems like you chose factor 5 for this study. If that is correct, I wonder why you chose 5 instead of 3, is this a wavelength consideration? I am not saying factor 5 is worse than 3, I just feel the authors need to elaborate a bit more, as this influences the discussion of the relative changes with respect to the sea ice thickness in Sect. 3.1.1.

>>>>>>>>>>>>>>>>>>>>>>>>>>>>>>>>>>>
Fixed: We will explain our choice of multiple for e-folding depth, but basically by three e-folding depths the light has reduced to ~5% of incidence and by five e-folding depths the light has reduced to ~0.7% of incidence. So whether we choose 3 or 5 e-folding depths is not too important.
<<<<<<<<<<<<<<<<<<<<<<<<<<<<<<<<<<<

P7L9: please also include a more recent reference for this statement, as this is the central motivation of this study.

>>>>>>>>>>>>>>>>>>>>>>>>>>>>>>>>>>>
Fixed: A more recent reference will be included.
<<<<<<<<<<<<<<<<<<<<<<<<<<<<<<<<<<<

P7L15: the second part of 3.1.2 needs rewriting, as it is very hard to follow how the authors describe Fig. 6b. In addition, what in the text is referred to as Figure 6 is actually Figure 5 (P7L9), whereas Figure 6b in the text should be Figure 6.

>>>>>>>>>>>>>>>>>>>>>>>>>>>>>>>>>>>
Fixed: Section 3.1.2 will be rewritten and figure numbers will be corrected
<<<<<<<<<<<<<<<<<<<<<<<<<<<<<<<<<<<

P8L22: 'however the intensity of peak increases with $\theta i$'. It is a bit difficult to follow this claim looking at Fig. 7, as at first glance the colors look the same, because the ranges of the color bars are not the same. I would suggest finding a different and consistent color bar. However, if the wide range of BRFs makes that too difficult, I think the authors should mention in the text already that the reader should pay attention to the varying ranges in the color bars of the respective sea ice types.

>>>>>>>>>>>>>>>>>>>>>>>>>>>>>>>>>>>
Fixed: The text will have a warning to observe the scale bars carefully in the figures. The reviewer understands the issue with the scale bars and range of values all too clearly and their suggestion is appreciated.
<<<<<<<<<<<<<<<<<<<<<<<<<<<<<<<<<<<

P9L6: 'However, the BRF does not decrease uniformly over the hemisphere with an increasing wavelength.' I suggest introducing Fig. 8 only after this sentence, to use the first sentences as an introduction and then describe the results of the figure. This would help the reading flow, as currently after introducing Fig. 8 the authors describe earlier results from other figures again.

>>>>>>>>>>>>>>>>>>>>>>>>>>>>>>>>>>>

Fixed: This change of order will be made as it improves the flow.
<<<<<<<<<<<<<<<<<<<<<<<<<<<<<<<<<<<<

P9L13: decreases by 13.94% compared to what?

>>>>>>>>>>>>>>>>>>>>>>>>>>>>>>>>>>>>
Fixed:  Changed to "...decreases by 14%  for the same change in wavelength."
<<<<<<<<<<<<<<<<<<<<<<<<<<<<<<<<<<<<

P11L1: Both Manninen et al. (2021) and Carlsen et al. (2020) study the influence of surface roughness on the BRDF of snow and the consequences for the calculation of the surface albedo and satellite retrievals, respectively. Manninen et al. (2021) from a modelling point of view, Carlsen et al. (2020) more from an observational side. Even though both studies investigate snow surfaces, the effects they are reporting are relevant for this study. However, the authors should elaborate more on how this relates to their results and give a bit more background when putting their study into perspective rather than just mentioning them. For example, the MODIS MCD43 product is never explained, and some readers might not know what it is.

>>>>>>>>>>>>>>>>>>>>>>>>>>>>>>>>>>>>
Fixed: Now reads "Although not bare sea ice, there is some benefit in the comparison with the effect of surface roughness on the BRF of snow. Manninen et al. (2021) modelled the BRF of snow and found surface roughness of snow increased the back scattering at large solar zenith angle. Carlsen et al. (2020) measured the HDRF of Antarctic snow surfaces in the wavelength band 490–585 nm using a 180 degree fish-eye camera in an airborne platform whilst retrieving the surface roughness using an airborne laser scanner and found that the backscatter is enhanced over rougher surfaces concluding that shadows and changing the effective angle of incidence were responsible. Accepting that snow and sea ice are different materials with some similar optical similarities the findings presented here are consistent with the works of Manninen et al (2021) and Carlsen et al. (2020).
<<<<<<<<<<<<<<<<<<<<<<<<<<<<<<<<<<<<

Section 4.3: Thanks for having made the model output available. It would be of interest for future users if you could mention at this point the increase in computational time necessary to increase the angular resolution of the simulations.

>>>>>>>>>>>>>>>>>>>>>>>>>>>>>>>>>>>>
Fixed: Doubling the model resolution in theta and phi would lead to 4 times as many directional quads and 16 times as many BRDF elements. However the model design employs efficiencies in the azimuthal rotational invariance of the volume scattering function in the medium, which means that the processing time scales with better efficiency than the number of BRDF elements. In practice run time for an increase in angular resolution f would be approx. f^3. The latter will be added to the manuscript.
<<<<<<<<<<<<<<<<<<<<<<<<<<<<<<<<<<<<

Section 4.3: Second paragraph about the intrinsic surface BRDF. This is as a first approximation true, however, the authors mentioned earlier themselves that the model also only considers direct and no diffuse illumination. However, the scene observed by the satellite is illuminated by both direct and diffuse radiation. So only propagating the surface BRF to the TOA is not entirely sufficient. Please also mention it again at this part of the manuscript.

>>>>>>>>>>>>>>>>>>>>>>>>>>>>>>>>>>>>
Fixed: The requested text, from the introduction, will be mentioned again in section 4.3
<<<<<<<<<<<<<<<<<<<<<<<<<<<<<<<<<<<<

P12L25: The entire study focuses on the reduction in uncertainties for the retrieval of e.g. albedo products from remote sensing. The reduction of uncertainties in global climate models comes a bit out of nowhere at this point. Please back that up with a more thorough explanation or leave it out as in my opinion the study does not need that additional motivation, especially as it seems a bit far-fetched in the way it is mentioned right now.

>>>>>>>>>>>>>>>>>>>>>>>>>>>>>>>>>>>>
Fixed: The reduction in the uncertainties in global climate models will be removed.
<<<<<<<<<<<<<<<<<<<<<<<<<<<<<<<<<<<<

Technical Corrections
P2L8: with

>>>>>>>>>>>>>>>>>>>>>>>>>>>>>>>>>>>>
Fixed: 'rwith' now 'with'.
<<<<<<<<<<<<<<<<<<<<<<<<<<<<<<<<<<<<

P2L18: sea ice (additional 'sea')

>>>>>>>>>>>>>>>>>>>>>>>>>>>>>>>>>>>>
Fixed: additional 'sea' removed.
<<<<<<<<<<<<<<<<<<<<<<<<<<<<<<<<<<<<

P2L22: snow kernels

>>>>>>>>>>>>>>>>>>>>>>>>>>>>>>>>>>>>
Fixed: 'kernal' now 'kernel'.
<<<<<<<<<<<<<<<<<<<<<<<<<<<<<<<<<<<<

P2L25: exist for sea ice

>>>>>>>>>>>>>>>>>>>>>>>>>>>>>>>>>>>>
Fixed: 'exit' now 'exists'.
<<<<<<<<<<<<<<<<<<<<<<<<<<<<<<<<<<<<

P4L4: ideal

>>>>>>>>>>>>>>>>>>>>>>>>>>>>>>>>>>>
Fixed: 'idea' now 'ideal'.
<<<<<<<<<<<<<<<<<<<<<<<<<<<<<<<<<<<

Fig. 1: phi symbols in text and figure not the same

>>>>>>>>>>>>>>>>>>>>>>>>>>>>>>>>>>>
Simply different fonts.
<<<<<<<<<<<<<<<<<<<<<<<<<<<<<<<<<<<

Fig. 2: color bar needs adjustments

>>>>>>>>>>>>>>>>>>>>>>>>>>>>>>>>>>>
The colour bar will be adjusted
<<<<<<<<<<<<<<<<<<<<<<<<<<<<<<<<<<<

Fig. 3: in the caption it says sigma = ... m, but sigma is a unitless quantity

>>>>>>>>>>>>>>>>>>>>>>>>>>>>>>>>>>>
Fixed: This change will be made throughout.
<<<<<<<<<<<<<<<<<<<<<<<<<<<<<<<<<<<

Figs. 5, 7 and 8: somehow the color bars are upside-down (including the labels)

>>>>>>>>>>>>>>>>>>>>>>>>>>>>>>>>>>>
Fixed: The numbers will be rotated by 90 degrees.
<<<<<<<<<<<<<<<<<<<<<<<<<<<<<<<<<<<

Fig. 7: the roughness values at the top are flipped, highest is now on the left and lowest on the right (as compared to the other Figures), BRF color bar is also flipped, and not the same throughout the Fig., thus it is hard to compare the different plots.

>>>>>>>>>>>>>>>>>>>>>>>>>>>>>>>>>>>
Fixed: The labels will be fixed.
<<<<<<<<<<<<<<<<<<<<<<<<<<<<<<<<<<<

P12L4: please define/explain HULIS

>>>>>>>>>>>>>>>>>>>>>>>>>>>>>>>>>>>
Fixed: text now reads "HUmic-Like Substances (HULIS)" and a reference to a review of HULIS will be given.
<<<<<<<<<<<<<<<<<<<<<<<<<<<<<<<<<<<

P12L14: please split that sentence up in two, it is currently very hard to read.

>>>>>>>>>>>>>>>>>>>>>>>>>>>>>>>>>>>
Fixed: Sentence will become two sentences.
<<<<<<<<<<<<<<<<<<<<<<<<<<<<<<<<<<<

P12L18: same as above, please split that sentence up in two.

\>\>\>\>\>\>\>\>\>\>\>\>\>\>\>\>\>\>\>\>\>\>\>\>\>\>\>\>\>\>\>\>\>\>\>\>\>\>\>\>

Fixed: Sentence will become two sentences.

<<<<<<<<<<<<<<<<<<<<<<<<<<<<<<<<<<<<<<<<

\>\>\>\>\>\>\>\>\>\>\>\>\>\>\>\>\>\>\>\>\>\>\>\>\>\>\>\>\>\>\>\>\>\>\>\>\>\>\>\>

Fixed: Sentence will become two sentences.

<<<<<<<<<<<<<<<<<<<<<<<<<<<<<<<<<<<<<<<<

---

## Author Comment (AC2)

Comment on tc-2021-366
Anonymous Referee #2

Referee comment on "The effects of surface roughness on the spectral (300–1400 nm) bidirectional reflectance distribution function (BRDF) of bare sea ice" by Maxim L. Lamare et al., The Cryosphere Discuss., https://doi.org/10.5194/tc-2021-366-RC2, 2022
The authors present a well-defined model study that investigates the effect of sea ice surface roughness on the directional reflectance of bare ice as a function of wavelength, sea ice thickness, and solar zenith angle. The work represents a further important contribution for the understanding of radiative transfer in a polar environment, even though the study shows its limitations. But these limitations are well addressed by the authors.

>>>>>>>>>>>>>>>>>>>>>>>>>>>>>>>>>>>>>
Thank you
<<<<<<<<<<<<<<<<<<<<<<<<<<<<<<<<<<<<<

General Comments
The study is about bare ice reflection properties. Therefore, I suggest to elaborate a little bit the meaning of this specific surface type in a meteorological context. In the current version, the introduction is mostly related to technical aspects. What is the seasonal contribution of bare ice in polar regions? Introduce the three ice types which are used in Sec. 2. Also give some more background on the surface roughness. What are typical scales?
An entire paragraph is listing snow surface studies without giving any details. However, in my opinion this part could be removed. The authors should focus on a review on bare ice studies dealing with surface reflection properties and their dependencies. A comparison with findings related to snow surfaces should be given in the discussion section.

>>>>>>>>>>>>>>>>>>>>>>>>>>>>>>>>>>>>>
Fixed: The introduction will be expanded to introduce the three bare ice types, the scales and surface roughness. The section on snow surface studies will be moved to the discussion and expanded.
<<<<<<<<<<<<<<<<<<<<<<<<<<<<<<<<<<<<<

I'm wondering if Section 3 could be restructured. In the current version it's no easy reading. Starting with the roughness and wavelength dependence might help (Fig. 8), without showing the nadir BRF plot (Fig. 4). The main message becomes clear from the contour plots already. Then show and discuss Fig. 5 (roughness and thickness). Based on Figs. 8 and 5 I would introduce the shift and broadening of the scattering peak by presenting Fig. 6. At the end show and discuss the roughness - SZA relation (Fig. 7).

>>>>>>>>>>>>>>>>>>>>>>>>>>>>>>>>>>>>>
Fixed: section 3 will be restructured following the outline suggested by referee 1.
<<<<<<<<<<<<<<<<<<<<<<<<<<<<<<<<<<<<<

Specific Comments

P1l12: "Different types of sea ice..." A lot of numbers quantifying the roughness effects are given at this part of the abstract. Try to reduce the information since it is a way too much.

>>>>>>>>>>>>>>>>>>>>>>>>>>>>>>>>>>>
Fixed: The amount of numerical information in the abstract will be reduced.
<<<<<<<<<<<<<<<<<<<<<<<<<<<<<<<<<<<

P2l2 and other: Omit the term "light" when you talk about solar radiation which is not within the visible wavelength range (400 – 700 nm)

>>>>>>>>>>>>>>>>>>>>>>>>>>>>>>>>>>>
Fixed. The term light will be removed when not discussing visible electromagnetic radiation, but UV and NIR electromagnetic radiation are still light.
<<<<<<<<<<<<<<<<<<<<<<<<<<<<<<<<<<<

P2l10: "The BRDF is a directional description of albedo..." The relation is more complex than stated here, be more precise. Also the HDRF should be introduced better than just saying that it is a proxy of the BRDF.

>>>>>>>>>>>>>>>>>>>>>>>>>>>>>>>>>>>
Fixed. Please see our reply to reviewer 1: A more in-depth discussion about biconical system will be given.
<<<<<<<<<<<<<<<<<<<<<<<<<<<<<<<<<<<

P2l33: "high sensitivity" – please elaborate

>>>>>>>>>>>>>>>>>>>>>>>>>>>>>>>>>>>
Fixed: A fuller explanation of what is sensitive to what will be given.
<<<<<<<<<<<<<<<<<<<<<<<<<<<<<<<<<<<

P3l2: "13 bands" – spectral bands, p3l7: "300 and 4000 nm" add wavelength afterwards, same as on p3l19 to make sure that wavelength is meant here

>>>>>>>>>>>>>>>>>>>>>>>>>>>>>>>>>>>
Fixed: These typos will be corrected.
<<<<<<<<<<<<<<<<<<<<<<<<<<<<<<<<<<<

P3l17: "... is required" Because it has not been done yet is not a convincing reason. What kind of consequences do you expect to derive from this study?

>>>>>>>>>>>>>>>>>>>>>>>>>>>>>>>>>>>
Fixed: Reasons will be included in the manuscript. We note here that the first reviewer gave excellent reasons: "This study marks an important contribution for the remote sensing community, as different retrieval products from aircraft and satellites rely on an accurate knowledge of the multiangular reflectance of sea ice, which so far has been underrepresented in the literature. The retrieval products in need for a better representation of the

surface reflectance anisotropy most notably include surface energy budget observations, but also include atmospheric retrievals that rely on surface reflectance corrections (e.g. cloud retrievals)."
<<<<<<<<<<<<<<<<<<<<<<<<<<<<<<<<<<<

Last paragraph of the introduction could be improved to reflect the outline in a better way.

>>>>>>>>>>>>>>>>>>>>>>>>>>>>>>>>>>>
Fixed: The last paragraph of the introduction will be reworded.
<<<<<<<<<<<<<<<<<<<<<<<<<<<<<<<<<<<

P3l32 / Fig. 1a: Do you really need Fig. 1a?

>>>>>>>>>>>>>>>>>>>>>>>>>>>>>>>>>>>
Figure 1a is a quick, clear, and unambiguous method to define the geometry and variables used in our paper. Whilst it is not absolutely essential it does add huge amounts of clarity to the work. Since no reason is given to remove it we have decided to keep it
<<<<<<<<<<<<<<<<<<<<<<<<<<<<<<<<<<<

Eq. (2) add "=pi*BRDF"

>>>>>>>>>>>>>>>>>>>>>>>>>>>>>>>>>>>
Fixed.
<<<<<<<<<<<<<<<<<<<<<<<<<<<<<<<<<<<

When I have understood correctly, the PlanarRad model was designed for aquatic radiative transfer. What justifies its use for calculating sea ice reflection properties? Is it a numerical model? Later (p5l9) Monte Carlo ray tracing is mentioned. Please explain a little more how this is related.

>>>>>>>>>>>>>>>>>>>>>>>>>>>>>>>>>>>
The description of the model will be reworded to be crystal clear which calculations are performed with the numerical model PlanarRad and which calculations use the ray tracing (roughness calculation). PanarRad was originally written for marine aquatic environments and sea-ice is a marine aquatic environment. The numerical part of the radiative model works in the same way as the model Hydrolight which is a very well-known model in hydrological optics.
<<<<<<<<<<<<<<<<<<<<<<<<<<<<<<<<<<<

P4l10: "calculation of the BRDF" Actually, the model allows rather the simulation of the biconical reflection. So it's not a real BR(D)F which is shown in the following plots.

>>>>>>>>>>>>>>>>>>>>>>>>>>>>>>>>>>>
Fixed: As outlined in the reply to review 1. A fuller more accurate description will be given of our biconical set-up.
<<<<<<<<<<<<<<<<<<<<<<<<<<<<<<<<<<<

P4l17+Fig. 1b and P4l28+Fig. 2: You could combine both figures as Fig 1a and Fig 1b. P5l3: Here the roughness parameter is without unit, but later and on p4l29 sigma is given in meters. Also in Fig. 3 its given without and with unit. Further, I suggest to move the definition of the roughness parameter to section 2.1.

>>>>>>>>>>>>>>>>>>>>>>>>>>>>>>>>>>>>
Fixed: The roughness parameter should not have units, this will be explained and corrected throughout the manuscript. The roughness definition will be moved as suggested. The figures will not be combined as this will create a weird right-angled figure and occupy more journal space.
<<<<<<<<<<<<<<<<<<<<<<<<<<<<<<<<<<<

P5l11: "using 10 rays per quad" Could you explain, how is this number selected?

>>>>>>>>>>>>>>>>>>>>>>>>>>>>>>>>>>>>
Fixed: the justification will be explained. It is a balance between reproducible simulation and computational effort. Further increasing the number of rays per quad did not increase the precision of the calculation.
<<<<<<<<<<<<<<<<<<<<<<<<<<<<<<<<<<<

P5l12: A scale height between 0.1 and 10 cm is chosen here. How do these numbers relate to real roughness features?

>>>>>>>>>>>>>>>>>>>>>>>>>>>>>>>>>>>>
Fixed: The roughness parameter should not have units. A detailed description of the following explanation will be given: Sigma is the elevation (height) standard deviation relative to the horizontal distance between two points on the surface and is unitless. Note the statement at page 5 line 4 is incorrect and will be removed "For example, if σ = 1, the slope between points located 1 mm apart has a standard deviation of 1 mm".
<<<<<<<<<<<<<<<<<<<<<<<<<<<<<<<<<<<

P5l14: The three sea ice types come somewhat out of the blue. Some more background should be already given in the introduction (see my first general comment).

>>>>>>>>>>>>>>>>>>>>>>>>>>>>>>>>>>>>
Fixed: Our justification for the three sea ices will be introduced and strengthed.
<<<<<<<<<<<<<<<<<<<<<<<<<<<<<<<<<<<

P5l19: "mass-ratio of 1 ng/g" Did you see any effect of black carbon for this low number? From Marks and King (2014), Figure 3, I don't expect any significant contribution.

>>>>>>>>>>>>>>>>>>>>>>>>>>>>>>>>>>>>
Fixed: A black carbon ratio of 1 ng g(-1) does has a small effect that is hard to discern on a graph.
<<<<<<<<<<<<<<<<<<<<<<<<<<<<<<<<<<<

P5l25: "increases by 29.5%...up to 630.7%" I would give only integer numbers (here and elsewhere)

>>>>>>>>>>>>>>>>>>>>>>>>>>>>>>>>>>>
Fixed: The precision of such numbers will be reduced.
<<<<<<<<<<<<<<<<<<<<<<<<<<<<<<<<<<<

P7l8: "...calculate the energy budget of the sea ice..." The sentence tries to motivate the direction of investigation. However, you should make this point already in the introduction, where the relation between satellite-based observations of the directional reflection, BRDF, BRF, ice albedo, and energy budget should be given.

>>>>>>>>>>>>>>>>>>>>>>>>>>>>>>>>>>>
Fixed: Referee 1 requested this argument be removed and it will be.
<<<<<<<<<<<<<<<<<<<<<<<<<<<<<<<<<<<

P7l32: "moves lower on the hemisphere" – maybe better say that it is shifted to higher viewing zenith angles

>>>>>>>>>>>>>>>>>>>>>>>>>>>>>>>>>>>
Fixed: The phrase will be reworded as suggested.
<<<<<<<<<<<<<<<<<<<<<<<<<<<<<<<<<<<

P9l27: "to a larger zenith angle" – better write "viewing zenith angle" here

>>>>>>>>>>>>>>>>>>>>>>>>>>>>>>>>>>>
Fixed: The phrase will be reworded as suggested.
<<<<<<<<<<<<<<<<<<<<<<<<<<<<<<<<<<<

P11l1: The comparison with findings for snow surfaces should be extended. Give also quantitative results.

>>>>>>>>>>>>>>>>>>>>>>>>>>>>>>>>>>>
Fixed: The section on snow surface studies will be moved to the discussion and expanded.
<<<<<<<<<<<<<<<<<<<<<<<<<<<<<<<<<<<

P11 Section 4.2: I would shift this subsection to the model setup section. Statements related to the surface in general should be part of the introduction.

>>>>>>>>>>>>>>>>>>>>>>>>>>>>>>>>>>>
Fixed: This text will be reworded and moved to the introduction.
<<<<<<<<<<<<<<<<<<<<<<<<<<<<<<<<<<<

Technical Corrections
P2l8: "rwith" – with

>>>>>>>>>>>>>>>>>>>>>>>>>>>>>>>>>>>

Fixed.
<<<<<<<<<<<<<<<<<<<<<<<<<<<<<<<<<<<

P2l12 and other: insert a space before references "roughness(e.g. Manninen, 1997)" –
roughness (e.g. Manninen, 1997)

>>>>>>>>>>>>>>>>>>>>>>>>>>>>>>>>>>
Fixed.
<<<<<<<<<<<<<<<<<<<<<<<<<<<<<<<<<<<

P2l22: "snow kernals" – kernels

>>>>>>>>>>>>>>>>>>>>>>>>>>>>>>>>>>
Fixed.
<<<<<<<<<<<<<<<<<<<<<<<<<<<<<<<<<<<<

P2l31: "isotopic" – isotropic; Do you mean a Lambertian assumption here?

>>>>>>>>>>>>>>>>>>>>>>>>>>>>>>>>>>
Fixed.
<<<<<<<<<<<<<<<<<<<<<<<<<<<<<<<<<<<<

P3l1 and other: "dependance" – dependence

>>>>>>>>>>>>>>>>>>>>>>>>>>>>>>>>>>
Fixed.
<<<<<<<<<<<<<<<<<<<<<<<<<<<<<<<<<<<<<

P4l4: "idea" – ideal

>>>>>>>>>>>>>>>>>>>>>>>>>>>>>>>>>>
Fixed.
<<<<<<<<<<<<<<<<<<<<<<<<<<<<<<<<<<<<<

Figure 3: Please increase the font size of the axis labeling.

>>>>>>>>>>>>>>>>>>>>>>>>>>>>>>>>>>
Fixed.
<<<<<<<<<<<<<<<<<<<<<<<<<<<<<<<<<<<<<

P6l3: "thickness" – ice thickness

>>>>>>>>>>>>>>>>>>>>>>>>>>>>>>>>>>
Fixed.
<<<<<<<<<<<<<<<<<<<<<<<<<<<<<<<<<<<<

Figure 4: Please increase the font size of the axis labeling.

>>>>>>>>>>>>>>>>>>>>>>>>>>>>>>>>>>

Fixed.
<<<<<<<<<<<<<<<<<<<<<<<<<<<<<<<<

P6l24: "… due to the large absorption in the ice dominating the signal …" Sounds strange.

>>>>>>>>>>>>>>>>>>>>>>>>>>>>>>>>
Fixed. Reworded.
<<<<<<<<<<<<<<<<<<<<<<<<<<<<<<<<

5: please check color bar labeling

>>>>>>>>>>>>>>>>>>>>>>>>>>>>>>>>
Fixed.
<<<<<<<<<<<<<<<<<<<<<<<<<<<<<<<<

7: roughness (sigma) is in wrong order, also check color bar labeling

>>>>>>>>>>>>>>>>>>>>>>>>>>>>>>>>
Fixed.
<<<<<<<<<<<<<<<<<<<<<<<<<<<<<<<<

8: roughness (sigma) is in wrong order, also check color bar labeling

>>>>>>>>>>>>>>>>>>>>>>>>>>>>>>>>
Fixed.
<<<<<<<<<<<<<<<<<<<<<<<<<<<<<<<<

P7l31: "as shown in Figure 6" – wrong reference

>>>>>>>>>>>>>>>>>>>>>>>>>>>>>>>>
Fixed.
<<<<<<<<<<<<<<<<<<<<<<<<<<<<<<<<

P8l6: "inter-dependant" – inter-dependent

>>>>>>>>>>>>>>>>>>>>>>>>>>>>>>>>
Fixed.
<<<<<<<<<<<<<<<<<<<<<<<<<<<<<<<<

P8l20: "inter-dependant" – inter-dependent

>>>>>>>>>>>>>>>>>>>>>>>>>>>>>>>>
Fixed.
<<<<<<<<<<<<<<<<<<<<<<<<<<<<<<<<

P10l12: Start a new paragraph with "Miao et al. (2020) …"

>>>>>>>>>>>>>>>>>>>>>>>>>>>>>>>>
Fixed.

<<<<<<<<<<<<<<<<<<<<<<<<<<<<<<<<

P10l27: "a airborne" – an airborne

>>>>>>>>>>>>>>>>>>>>>>>>>>>>>>
Fixed.
<<<<<<<<<<<<<<<<<<<<<<<<<<<<<<<<

---

## Author Response (AR1)

Reply to Comment on tc-2021-366

Anonymous Referee #1
Referee comment on "The effects of surface roughness on the spectral (300–1400 nm) bidirectional reflectance distribution function (BRDF) of bare sea ice" by Maxim L. Lamare et al., The Cryosphere Discuss., https://doi.org/10.5194/tc-2021-366-RC1, 2022
Within the manuscript, the influence of the surface roughness on the reflectance anisotropy of bare sea ice is simulated for different wavelengths, illumination conditions, and sea ice thicknesses. For this purpose, the authors employ the radiative transfer model PlanarRad. From this, the size and shape of the forward scattering maximum is discussed for different types of sea ice (first-year, multi-year, melting sea ice).

This study marks an important contribution for the remote sensing community, as different retrieval products from aircraft and satellites rely on an accurate knowledge of the multiangular reflectance of sea ice, which so far has been underrepresented in the literature. The retrieval products in need for a better representation of the surface reflectance anisotropy most notably include surface energy budget observations, but also include atmospheric retrievals that rely on surface reflectance corrections (e.g. cloud retrievals).

The manuscript is concise and the figures are generally of a good quality. However, there are some aspects that need further attention in my opinion. After some general comments, the more specific comments and suggestions for technical corrections follow below.

>>>>>>>>>>>>>>>>>>>>>>>>>>>>>>>>>>>>>>
Thank you.
<<<<<<<<<<<<<<<<<<<<<<<<<<<<<<<<<<<<<

General Comments
(1) Nomenclature of reflectance quantities: The manuscript is missing a rigorous and consistent usage of reflectance terms. Even though BRDF and BRF are defined in Section 2.1, the usage throughout the text is inconsistent and, in some cases, erroneous. For example, the title indicates a study focused on the BRDF. Instead, due to the limited angular resolution in the viewing geometry, directional-conical results are presented (compare Schaepman-Strub et al., 2006). Other instances include, e.g. P2L19 (BRF stands for Bidirectional reflectance factor, and BRF is not really an approximation of BRDF, it relates the reflected radiant flux of a sample surface to the radiant flux of an ideal Lambertian surface irradiated under the same conditions, as you define yourself in Sect. 2.1), P2L28 and P3L2 (the BRDF can never be measured), and P2L31(what is an isotropic albedo?). I understand that using BRDF is somewhat established in the literature, however I think the authors need to be more careful and follow the recommendations put forward by Schaepman-Strub et al. (2006) to make an effort to improve the usage of reflectance terms in the literature. This is not limited to the above-mentioned cases, and I suggest the authors check again their usage of terms throughout the manuscript (including the introduction when discussing other studies). You also need to mention that you are not simulating a BRF, but an approximation. If you call it a BRF you are assuming the reflected radiance is constant throughout the viewing quads.

>>>>>>>>>>>>>>>>>>>>>>>>>>>>>>>>>>>>>>
We agree with the reviewer that paper could be more precise in the definition of the properties evaluated, and we have added an initial clarification of this and modify the manuscript for consistency. The model used is designed in the

same way as the model HydroLight which is well-known in the field of hydrological optics. These models work with "quad-averaged radiances" where it assumed the radiance is constant over the solid angle subtended by each segment of the angular discretisation, both for "input" and "output". Therefore the BRDFs evaluated are biconical (geometry equivalent to case 5 in Table 2 of Scheapman-Strub et al. 2006), we have indicated this in the manuscript with extra text in methodology to make it very clear.

"Radiance is constant over the solid angle subtended by each segment of the angular discretisation, both for upwelling and downwelling and the directional reflectance is evaluated as conical-conical or biconical (geometry equivalent to case 5 in Table 2 of Schaepman-Strub et al. (2006)."

P2L19 - "Bidirectional Reflection Function (BRF) as an approximation of BRDF" will be reworded as "Bidirectional Reflection Function (BRF) as an alternative to the BRDF".

References to "BRDF" and "BRF" have been corrected to "CCRF" or "conical-conical reflectance factor" or similar. The title of the manuscript has also been changed:-

"The effects of surface roughness on the calculated spectral (300 – 1400 nm) conical-conical reflectance factor (CCRF), as an alternative to bidirectional reflectance distribution function (BRDF), of bare sea ice "

And the following text has been added earlier on.

"Strictly the quantity, BRDF, cannot be measured and often other directional reflectance measurements are undertaken as an alternative or approximation of BRDF (Schaepman-Strub et al., 2006). There is a large number of terms in the literature for quantities that are measurable alternatives to BRDF that may not have been used uniformly and will be described herein as directional reflectance"
<<<<<<<<<<<<<<<<<<<<<<<<<<<<<<<<<<

(2) Introduction: In my opinion, the authors need to elaborate more on the references used in the Introduction. Long lists of references are given for a specific point of discussion, but simply mentioning the reference is not enough. For example, not just mention wavelength (....), but actually mention what the current state of the art is regarding wavelength-dependence of the BRDF/HDRF. So far, the introduction only mentions that different effects on the reflectance anisotropy have been measured before, but these effects are not described at all yet. For a proper state-of-the-art overview, more details need to be given already here. This will also help later to put the results of this study in perspective.

>>>>>>>>>>>>>>>>>>>>>>>>>>>>>>>>>>>
This section was removed as per the second reviewer request and the introduction reads much better. It was never our intention to provide a state-of-the-art review.
<<<<<<<<<<<<<<<<<<<<<<<<<<<<<<<<<<<<

(3) Results: The structure of the results section needs adjustments, as currently the storyline is hard to follow as effects of roughness, thickness, wavelength and solar zenith angle are mixed throughout the different subsections. In addition, the separation of the Nadir BRF results complicates things in my eyes, as this could be discussed together with the 2D BRF. My suggestion would be: 3.1 Roughness and sea ice thickness (showing Figures 4-6, but maybe even switching the order of Figures 5 and 4), 3.2 Roughness and solar zenith angle (Fig. 7), 3.3 Roughness and wavelength (Fig. 8). That would make the structure easier to follow.

>>>>>>>>>>>>>>>>>>>>>>>>>>>>>>>>>>>>>
Fixed: The structure of the results section is now
3.1 roughness and sea ice thickness
32. roughness and solar zenith angle
3.3 roughness and wavelength
<<<<<<<<<<<<<<<<<<<<<<<<<<<<<<<<<<<<

(4) Related to the last comment, Section 3.2.1 starts describing Figure 5 again as if it is mentioned for the first time in the text. I recommend to restructure and discuss the effect of the thickness already together with the effect of the roughness. First half of 3.2.1 actually discusses roughness again from earlier as well. The influence of the sea ice thickness shows the influence of the underlying surface, i.e. the ocean BRF. Thus, the values are lower as you clearly demonstrate. However, I think it is worth noting in the text that the shape of the BRF itself remains unaltered, meaning the shape and size of the forward scattering peak seems similar for all sea ice thicknesses independent of the sea ice type.

>>>>>>>>>>>>>>>>>>>>>>>>>>>>>>>>>>>>>
Fixed: The results section has been restructured, and the latter is explicitly stated in the conclusions "For small amounts of surface roughness, a reduction in sea ice thickness decreases the quasi-isotropic part of the CCRF, affecting the forward scattering peak very little." And "For small roughnesses ($\sigma < 0.01$ ) the quasi-isotropic part of the CCRF is affected by a changing thickness and for large roughnesses ($\sigma \geqslant 0.01$) the forward scattering peak is also affected."

<<<<<<<<<<<<<<<<<<<<<<<<<<<<<<<<<<<<

(5) The authors use the word 'quasi-infinite' in the figures, but 'optically thick' throughout the text. If the more commonly used term 'semi-infinite' would be used consistently, it would make the text much easier to read, and also put the text more in line with the Lamare et al. (2016) terminology. If the authors had any specific reason to name it differently, this should be emphasized more.

>>>>>>>>>>>>>>>>>>>>>>>>>>>>>>>>>>>>>
Fixed: The authors have moved away from the term 'semi-infinite' as it is jargon and non-sensical. We have corrected the figures to report 'optically thick' as opposed to 'quasi-infinite' to be more in keeping with the text.
<<<<<<<<<<<<<<<<<<<<<<<<<<<<<<<<<<<<

(6) I agree with the authors that studying the semi-infinite (optically thick) sea ice is important to understand the intrinsic surface BRDF of bare sea ice. However, I feel the study could benefit from including a figure looking at the spectral and solar zenith angle dependence of the BRF for another sea ice thickness that is closer to natural sea ice (as melting sea ice with a thickness of 20 m is more a theoretical consideration). I believe seeing the effects of wavelength and illumination angle on both the theoretical (semi- infinite) and the more realistic (e.g. 50 cm/100 cm) thicknesses would be of interest to many readers. Adding an additional figure could work (maybe restricting to only one roughness value at the other thicknesses), or maybe also adding another column for an additional sea ice thickness in Figs. 7 and 8 could be an option.

>>>>>>>>>>>>>>>>>>>>>>>>>>>>>>>>>>>>
The data presented in figure 5 for BRF with different thicknesses of sea ice other than optically thick/quasi-infinite thickness would meet this request in part. The paper already has a lot of large figures, and it would take over 100s figures to present all the possible degrees of freedom (solar zenith angle, wavelength, surface roughness and thickness) that could be requested of the dataset. All the data is freely available at doi:10.5281/zenodo.5733402 and any reader can download these data and plot the variations they wish to plot. We wish to add the entire focus of the paper is to explore the effect of surface roughness on the reflectivity of sea ice, not the effect of other variables at constant surface roughness.
<<<<<<<<<<<<<<<<<<<<<<<<<<<<<<<<<<<<

(7) Section 4.2: This is a very important section that puts the choice of the modeled roughness parameter in perspective. However, this should be included in the Methods section already, when mentioning the range of modeled roughness parameters at the end of Sect. 2.2. I also suggest the authors consider making the roughness considerations a separate subsection within the methods, e.g. after 2.2 Model description. As this is a vital part of the study, it should have a separate section in the Methods, and the choice of roughness parameters needs to be motivated already at this point. At the moment, 4.2 seems a bit out of place in the discussion.

>>>>>>>>>>>>>>>>>>>>>>>>>>>>>>>>>>>>
Fixed: We have added the roughness section to the methods. This was a good idea.
<<<<<<<<<<<<<<<<<<<<<<<<<<<<<<<<<<<<

(8) Please be consistent about the usage of the roughness parameter sigma throughout the manuscript. You first state it is unitless, but then give 'm' as a unit on several occasions, e.g. captions of Figs. 3 and 4, or on P6L23.

>>>>>>>>>>>>>>>>>>>>>>>>>>>>>>>>>>>>
Fixed: The roughness parameter should not have units. The following example is now given to help

"For example, if σ = 0.5, two points located 2mm apart would have their heights drawn from a normal distribution of mean zero and standard deviation of 1mm."

And with a helpful diagram to help visualise.

"In the work presented here, 5 modelled surfaces were generated with an elevation standard deviation, σ = 0.001, 0.005, 0.01, 0.05 and 0.1 (Figure XX). The surfaces were generated using 10 rays per quad (4320 rays in total) with results averaged over 2000 surfaces. The roughness model being scale invariant, and the relative amplitude defined as 1 meter, the scale height of the roughness is 0.1, 0.5, 1, 5 and 10 cm. "

Sigma is the elevation (height) standard deviation relative to the horizontal distance between two points on the surface and is unitless.

In the new section 2.3 the following text is introduced to explain that the roughness chosen were in keeping with the work of Tucker et al. (2013).

"To cover a wide range of conditions, a selection of five surface roughness parameters, defined by the standard deviation of the height of the surface were picked, with a standard deviation of 1 mm to 10 cm relative to two surface points 1m horizontally apart . The range of surface roughness is in agreement with observations reported in the literature for small scale roughnesses (e.g. Tucker et al., 2013)."
<<<<<<<<<<<<<<<<<<<<<<<<<<<<<<<<<<

(9) When you mention specific values of the BRF or changes in %, you give values with 2 or even 3 decimal places, which is an accuracy not needed (with respect to the mentioned model limitations) and makes reading the numbers quite cumbersome. In my view, rounding the values to integer numbers seems to be more than sufficient.

>>>>>>>>>>>>>>>>>>>>>>>>>>>>>>>>>>>>
Fixed: The precision is now smaller and consistent throughout the paper.
<<<<<<<<<<<<<<<<<<<<<<<<<<<<<<<<<<<

Specific Comments
Title: In addition to using the correct reflectance term in the title (see general comments above), I suggest to include modeled/simulated in the title as well so that the reader immediately knows what to expect from the study.

>>>>>>>>>>>>>>>>>>>>>>>>>>>>>>>>>>>>
The title now read
"The effects of surface roughness on the calculated spectral (300 – 1400 nm) conical-conical reflectance factor (CCRF) as an alternative to bidirectional reflectance distribution function (BRDF) of bare sea ice"
<<<<<<<<<<<<<<<<<<<<<<<<<<<<<<<<<<<

Abstract: the mentioning of quads is not clear as this is not a commonly known term. You either need to define it or express the point you want to make differently. In addition, the angular resolution should be mentioned directly.

>>>>>>>>>>>>>>>>>>>>>>>>>>>>>>>>>>>>>>
Fixed: The following text has been added to the abstract

"The hemisphere was split in to 216 quadrangular regions or quads."
<<<<<<<<<<<<<<<<<<<<<<<<<<<<<<<<<<<<

P2L8: please be more accurate with the definition of BRDF at this first instance, it does not describe the relation between illumination and viewing angles, but of the incident and reflected radiation of all sets of illumination and viewing angles.

>>>>>>>>>>>>>>>>>>>>>>>>>>>>>>>>>>>>>>
Fixed: The original text.

"The Bidirectional Reflectance Distribution Function (BRDF) is a directional description of albedo, describing the relationship between illumination and viewing angles"

Was replaced with

"The Bidirectional reflectance Distribution Functions (BRDF) is a derivative distribution function that maps its contribution of incident irradiance from a direction to the reflected radiance in another direction"

<<<<<<<<<<<<<<<<<<<<<<<<<<<<<<<<<<<<

P2L10: add Schaepman-Strub et al. (2006) reference already at this point

>>>>>>>>>>>>>>>>>>>>>>>>>>>>>>>>>>>>>>
Done
<<<<<<<<<<<<<<<<<<<<<<<<<<<<<<<<<<<<

P3L14: please avoid statements of novelty in that way

>>>>>>>>>>>>>>>>>>>>>>>>>>>>>>>>>>>>>>
Fixed: The statement has been removed
<<<<<<<<<<<<<<<<<<<<<<<<<<<<<<<<<<<<

P6L17: add 'between ... nm wavelength', as otherwise it could sound like these are the chosen ice thickness values

>>>>>>>>>>>>>>>>>>>>>>>>>>>>>>>>>>>>>>
Fixed:  This has been removed to clarify we are not talking about nm thicknesses of sea ice.
<<<<<<<<<<<<<<<<<<<<<<<<<<<<<<<<<<<<

Fig. 4: the different lines are hard to distinguish, especially for the different roughness values for the quasi-infinite cases

>>>>>>>>>>>>>>>>>>>>>>>>>>>>>>>>>>>>>>>
Excellent – this is what Figure 4 is demonstrating. The following note has been added in the figure caption to this effect.

"The changes in nadir CCRF owing to changes roughness are hard to discern, especially relative to changes in thickness."
<<<<<<<<<<<<<<<<<<<<<<<<<<<<<<<<<<<

Sect. 3.1.1, strongest relative change in nadir BRF for melting sea ice. As far as I understand it, that is to be expected due to the much larger semi-infinite sea ice thickness of about 20 m. I think the authors should elaborate a bit on this and include more details on how these quantities were calculated in Lamare et al. (2016). P6L8 indicates that you calculated the e-folding depth times 3 or 5. If I compare to Table 1 in Lamare et al. (2016), it seems like you chose factor 5 for this study. If that is correct, I wonder why you chose 5 instead of 3, is this a wavelength consideration? I am not saying factor 5 is worse than 3, I just feel the authors need to elaborate a bit more, as this influences the discussion of the relative changes with respect to the sea ice thickness in Sect. 3.1.1.

>>>>>>>>>>>>>>>>>>>>>>>>>>>>>>>>>>>>>
Fixed: Section 3.1.1 no longer exists as reviewer requested earlier and the text has been changed woven into another section. The following text has been added

"Previously sea ice was considered to be optically thick at 3 $e$-folding depths, *i.e.* where over 95% of diffuse incident light is attenuated (France et al., 2011), in the work described here 5 e-folding depths *i.e.* where over 99% of diffuse incident light is attenuated, was used as an over cautious approach because unlike previous studies, Lamare et al. (2016); Marks and King (2014, 2013); Redmond Roche and King (2022), the study described here was using direct, not diffuse radiation. King et al. (2005) demonstrate that the decay of direct illumination in the near surface region of sea ice is not asymptotic."
<<<<<<<<<<<<<<<<<<<<<<<<<<<<<<<<<<<<

P7L9: please also include a more recent reference for this statement, as this is the central motivation of this study.

>>>>>>>>>>>>>>>>>>>>>>>>>>>>>>>>>>>>
Fixed: Two more, recent, references added.
<<<<<<<<<<<<<<<<<<<<<<<<<<<<<<<<<<

P7L15: the second part of 3.1.2 needs rewriting, as it is very hard to follow how the authors describe Fig. 6b. In addition, what in the text is referred to as Figure 6 is actually Figure 5 (P7L9), whereas Figure 6b in the text should be Figure 6.

>>>>>>>>>>>>>>>>>>>>>>>>>>>>>>>>>>>>

Fixed: Section 3.1.2 no longer exists and the text describing the figure has been incorporated into two different sections as per the reviewer's original instructions. The figures have been re-ordered.
<<<<<<<<<<<<<<<<<<<<<<<<<<<<<<<<<<<<

P8L22: 'however the intensity of peak increases with θi'. It is a bit difficult to follow this claim looking at Fig. 7, as at first glance the colors look the same, because the ranges of the color bars are not the same. I would suggest finding a different and consistent color bar. However, if the wide range of BRFs makes that too difficult, I think the authors should mention in the text already that the reader should pay attention to the varying ranges in the color bars of the respective sea ice types.

>>>>>>>>>>>>>>>>>>>>>>>>>>>>>>>>>>>>>
Fixed: The reviewer is correct with the issues with colour bars. The caption to the original figure 7 did have the following text "Note that the scale of the colour bar varies for the different illumination angles in order to visualise clearly the BRF pattern". The same text has been included as a warning to observe the scale bars carefully in the main text.
<<<<<<<<<<<<<<<<<<<<<<<<<<<<<<<<<<<<

P9L6: 'However, the BRF does not decrease uniformly over the hemisphere with an increasing wavelength.' I suggest introducing Fig. 8 only after this sentence, to use the first sentences as an introduction and then describe the results of the figure. This would help the reading flow, as currently after introducing Fig. 8 the authors describe earlier results from other figures again.

>>>>>>>>>>>>>>>>>>>>>>>>>>>>>>>>>>>>>
Fixed: The sentence "However, the BRF does not decrease uniformly over the hemisphere with an increasing wavelength" now appears before any mention of figure 8.
<<<<<<<<<<<<<<<<<<<<<<<<<<<<<<<<<<<<

P9L13: decreases by 13.94% compared to what?

>>>>>>>>>>>>>>>>>>>>>>>>>>>>>>>>>>>>>
Fixed:  The sentence is confusing and has been removed.
<<<<<<<<<<<<<<<<<<<<<<<<<<<<<<<<<<<<

P11L1: Both Manninen et al. (2021) and Carlsen et al. (2020) study the influence of surface roughness on the BRDF of snow and the consequences for the calculation of the surface albedo and satellite retrievals, respectively. Manninen et al. (2021) from a modelling point of view, Carlsen et al. (2020) more from an observational side. Even though both studies investigate snow surfaces, the effects they are reporting are relevant for this study. However, the authors should elaborate more on how this relates to their results and give a bit more background when putting their study into perspective rather than just mentioning them. For example, the MODIS MCD43 product is never explained, and some readers might not know what it is.

>>>>>>>>>>>>>>>>>>>>>>>>>>>>>>>>>>>>>

Fixed: Now reads "Although not bare sea ice, there is some benefit in the comparison with the effect of surface roughness on the BRF of snow. Manninen et al. (2021) modelled the BRF of snow and found surface roughness of snow increased the back scattering at large solar zenith angle. Carlsen et al. (2020) measured the HDRF of Antarctic snow surfaces in the wavelength band 490–585 nm using a 180 degree fish-eye camera in an airborne platform whilst retrieving the surface roughness using an airborne laser scanner and found that the backscatter is enhanced over rougher surfaces concluding that shadows and changing the effective angle of incidence were responsible. Accepting that snow and sea ice are different materials with some similar optical similarities the findings presented here are consistent with the works of Manninen et al (2021) and Carlsen et al. (2020).
<<<<<<<<<<<<<<<<<<<<<<<<<<<<<<<<<<<<

Section 4.3: Thanks for having made the model output available. It would be of interest for future users if you could mention at this point the increase in computational time necessary to increase the angular resolution of the simulations.

>>>>>>>>>>>>>>>>>>>>>>>>>>>>>>>>>>>>
Fixed: Doubling the model resolution in theta and phi would lead to 4 times as many directional quads and 16 times as many BRDF elements. However the model design employs efficiencies in the azimuthal rotational invariance of the volume scattering function in the medium, which means that the processing time scales with better efficiency than the number of BRDF elements. In practice run time for an increase in angular resolution f would be approx. f^3. The latter was added to the manuscript as

"To note that computational time scales roughly as $f^3$ where f is the angular resolution."
<<<<<<<<<<<<<<<<<<<<<<<<<<<<<<<<<<<<

Section 4.3: Second paragraph about the intrinsic surface BRDF. This is as a first approximation true, however, the authors mentioned earlier themselves that the model also only considers direct and no diffuse illumination. However, the scene observed by the satellite is illuminated by both direct and diffuse radiation. So only propagating the surface BRF to the TOA is not entirely sufficient. Please also mention it again at this part of the manuscript.

>>>>>>>>>>>>>>>>>>>>>>>>>>>>>>>>>>>>
Fixed: The text has been altered to include mention of direct and diffuse illumination.
<<<<<<<<<<<<<<<<<<<<<<<<<<<<<<<<<<<<

P12L25: The entire study focuses on the reduction in uncertainties for the retrieval of e.g. albedo products from remote sensing. The reduction of uncertainties in global climate models comes a bit out of nowhere at this point. Please back that up with a more thorough explanation or leave it out as in my opinion the study does not need that additional motivation, especially as it seems a bit far-fetched in the way it is mentioned right now.

>>>>>>>>>>>>>>>>>>>>>>>>>>>>>>>>>>>
Fixed: The text about the reduction in the uncertainties in global climate models has been removed.
<<<<<<<<<<<<<<<<<<<<<<<<<<<<<<<<<<<

Technical Corrections
P2L8: with

>>>>>>>>>>>>>>>>>>>>>>>>>>>>>>>>>>>
Fixed: 'rwith' now 'with'.
<<<<<<<<<<<<<<<<<<<<<<<<<<<<<<<<<<<

P2L18: sea ice (additional 'sea')

>>>>>>>>>>>>>>>>>>>>>>>>>>>>>>>>>>>
Fixed: additional 'sea' removed.
<<<<<<<<<<<<<<<<<<<<<<<<<<<<<<<<<<<

P2L22: snow kernels

>>>>>>>>>>>>>>>>>>>>>>>>>>>>>>>>>>>
Fixed: 'kernal' now 'kernel'.
<<<<<<<<<<<<<<<<<<<<<<<<<<<<<<<<<<<

P2L25: exist for sea ice

>>>>>>>>>>>>>>>>>>>>>>>>>>>>>>>>>>>
Fixed: 'exit' now 'exists'.
<<<<<<<<<<<<<<<<<<<<<<<<<<<<<<<<<<<

P4L4: ideal

>>>>>>>>>>>>>>>>>>>>>>>>>>>>>>>>>>>
Fixed: 'idea' now 'ideal'.
<<<<<<<<<<<<<<<<<<<<<<<<<<<<<<<<<<<

Fig. 1: phi symbols in text and figure not the same

>>>>>>>>>>>>>>>>>>>>>>>>>>>>>>>>>>>
Simply different fonts.
<<<<<<<<<<<<<<<<<<<<<<<<<<<<<<<<<<<

Fig. 2: color bar needs adjustments

>>>>>>>>>>>>>>>>>>>>>>>>>>>>>>>>>>>
The colour bar will be adjusted
<<<<<<<<<<<<<<<<<<<<<<<<<<<<<<<<<<<

Fig. 3: in the caption it says sigma = ... m, but sigma is a unitless quantity

>>>>>>>>>>>>>>>>>>>>>>>>>>>>>>>>>>>>
Fixed: Units have been removed from sigma throughout.
<<<<<<<<<<<<<<<<<<<<<<<<<<<<<<<<<<

Figs. 5, 7 and 8: somehow the color bars are upside-down (including the labels)

>>>>>>>>>>>>>>>>>>>>>>>>>>>>>>>>>>>>
Fixed: The labels on the colour bars have been altered.
<<<<<<<<<<<<<<<<<<<<<<<<<<<<<<<<<<<

Fig. 7: the roughness values at the top are flipped, highest is now on the left and lowest on the right (as compared to the other Figures), BRF color bar is also flipped, and not the same throughout the Fig., thus it is hard to compare the different plots.

>>>>>>>>>>>>>>>>>>>>>>>>>>>>>>>>>>>>
Fixed: The labels have been fixed.
<<<<<<<<<<<<<<<<<<<<<<<<<<<<<<<<<<<<

P12L4: please define/explain HULIS

>>>>>>>>>>>>>>>>>>>>>>>>>>>>>>>>>>>>
Fixed: text now reads "HUmic-Like Substances (HULIS)" and references to HULIS in snow and ice are given: Beine e al 2012,France et al 2012,Voisin et al 2012.
<<<<<<<<<<<<<<<<<<<<<<<<<<<<<<<<<<<<

P12L14: please split that sentence up in two, it is currently very hard to read.

>>>>>>>>>>>>>>>>>>>>>>>>>>>>>>>>>>>>
Fixed: Sentence is now three sentences.
<<<<<<<<<<<<<<<<<<<<<<<<<<<<<<<<<<<<

P12L18: same as above, please split that sentence up in two.

>>>>>>>>>>>>>>>>>>>>>>>>>>>>>>>>>>>>
Fixed: Sentence is now three sentences.
<<<<<<<<<<<<<<<<<<<<<<<<<<<<<<<<<<<<

Comment on tc-2021-366
Anonymous Referee #2

Referee comment on "The effects of surface roughness on the spectral (300–1400 nm)
bidirectional reflectance distribution function (BRDF) of bare sea ice" by Maxim L. Lamare et
al., The Cryosphere Discuss., https://doi.org/10.5194/tc-2021-366-RC2, 2022
The authors present a well-defined model study that investigates the effect of sea ice
surface roughness on the directional reflectance of bare ice as a function of wavelength, sea
ice thickness, and solar zenith angle. The work represents a further important contribution
for the understanding of radiative transfer in a polar environment, even though the study
shows its limitations. But these limitations are well addressed by the authors.

>>>>>>>>>>>>>>>>>>>>>>>>>>>>>>>>>>>>>>
Thank you
<<<<<<<<<<<<<<<<<<<<<<<<<<<<<<<<<<<<<

General Comments
The study is about bare ice reflection properties. Therefore, I suggest to elaborate a little bit
the meaning of this specific surface type in a meteorological context. In the current version,
the introduction is mostly related to technical aspects. What is the seasonal contribution of
bare ice in polar regions? Introduce the three ice types which are used in Sec. 2. Also give
some more background on the surface roughness. What are typical scales?
An entire paragraph is listing snow surface studies without giving any details. However, in
my opinion this part could be removed. The authors should focus on a review on bare ice
studies dealing with surface reflection properties and their dependencies. A comparison
with findings related to snow surfaces should be given in the discussion section.

>>>>>>>>>>>>>>>>>>>>>>>>>>>>>>>>>>>>>>
Fixed: The snow material has been removed and remaining text deals with
bare ice studies only. Typical scales are now given in the method section
specifically

"Sea ice roughness shows significant spatial variability, with vertical features
ranging from the millimetre-scale to the meter scale (e.g. Manninen, 1997;
Peterson et al., 2008). The larger surface roughness features are generally
caused by the deformation of the sea ice, forming rubble fields and pressure
ridges that can reach 10 to 20 m in height (Tucker et al., 2013). At a smaller
scale, brash ice, ridged blocks or frost flowers can create roughness with a
standard deviation of a few millimetres to centimetres. To cover a wide range
of conditions, a selection of five surface roughness parameters, defined by the
standard deviation of the height of the surface were picked, with a standard
deviation of 1 mm to 10 cm relative to two surface points 1m horizontally
apart . The range of surface roughness is in agreement with observations
reported in the literature for small scale roughnesses (e.g. Tucker
et al., 2013). Random surface realisations were generated to calculate the
surface roughness in the model, which is rotationally invariant.

Therefore, Planarrad produces a random surface roughness, that has no specific structure or pattern. Specific complicated shapes present in sea ice, such as pressure ridges were not modelled."

The following text is included in the introduction:

"In this work, the radiative-transfer model PlanarRad (Hedley, 2008, 2015) was used to model the CCRF (conical-conical reflectance factor) of three different types of sea ice from 300 to 1400 nm with varying thicknesses as a function of surface roughness in two steps. Firstly, the BRDF of three different types of sea ice with a thickness large enough to be optically thick was modelled with an increasing surface roughness. Secondly, the calculations performed in the first step were repeated, but the optically thick thicknesses were replaced with fixed thicknesses of 50 cm and 100 cm for each type of sea ice. The optical properties of the three types of bare sea ice are chosen to represent multiyear sea ice, first year sea ice and melting sea ice and will be described in detail in the methodology".

<<<<<<<<<<<<<<<<<<<<<<<<<<<<<<<<<<<<<

I'm wondering if Section 3 could be restructured. In the current version it's no easy reading. Starting with the roughness and wavelength dependence might help (Fig. 8), without showing the nadir BRF plot (Fig. 4). The main message becomes clear from the contour plots already. Then show and discuss Fig. 5 (roughness and thickness). Based on Figs. 8 and 5 I would introduce the shift and broadening of the scattering peak by presenting Fig. 6. At the end show and discuss the roughness - SZA relation (Fig. 7).

>>>>>>>>>>>>>>>>>>>>>>>>>>>>>>>>>>>>>
Fixed: section 3 has been completely restructured following the outline suggested by the other referee:-
3.1 roughness and sea ice thickness
32. roughness and solar zenith angle
3.3 roughness and wavelength
<<<<<<<<<<<<<<<<<<<<<<<<<<<<<<<<<<<<

Specific Comments
P1l12: "Different types of sea ice..." A lot of numbers quantifying the roughness effects are given at this part of the abstract. Try to reduce the information since it is a way too much.

>>>>>>>>>>>>>>>>>>>>>>>>>>>>>>>>>>>>>
Fixed: The amount of numerical information in the abstract has been reduced. The three sentences that describe the specific example have been removed.
<<<<<<<<<<<<<<<<<<<<<<<<<<<<<<<<<<<<

P2l2 and other: Omit the term "light" when you talk about solar radiation which is not within the visible wavelength range (400 – 700 nm)

>>>>>>>>>>>>>>>>>>>>>>>>>>>>>>>>>>>>>

Fixed. The term light will be removed when not discussing visible electromagnetic radiation, and yet UV and NIR electromagnetic radiation are still light.
<<<<<<<<<<<<<<<<<<<<<<<<<<<<<<<<<<<<

P2l10: "The BRDF is a directional description of albedo…" The relation is more complex than stated here, be more precise. Also the HDRF should be introduced better than just saying that it is a proxy of the BRDF.

>>>>>>>>>>>>>>>>>>>>>>>>>>>>>>>>>>>>
Fixed. BRDF is now described as
"The Bidirectional Reflectance Distribution Functions (BRDF) is a derivative distribution function that maps its contribution of incident irradiance from a direction to the reflected radiance in another direction". HDRF is a described as a field measurable alternative to BRDF. Repeated reference is made to the paper of Schaepman-Strub et al. 2006 where these quantities are defined in detail that does not need repeating here.)
<<<<<<<<<<<<<<<<<<<<<<<<<<<<<<<<<<<<

P2l33: "high sensitivity" – please elaborate

>>>>>>>>>>>>>>>>>>>>>>>>>>>>>>>>>>>>
Fixed: In their conclusions Jin and Simpson (1999) state "The reflectance anisotropy is also sensitive to the scattering characteristics of snow and the roughness of the sea ice surface". Thus the phrase will be edited to "showed that sea ice has a larger reflectance anisotropy in the forward observation direction and is sensitive to solar elevation and surface roughness."
<<<<<<<<<<<<<<<<<<<<<<<<<<<<<<<<<<<<

P3l2: "13 bands" – spectral bands, p3l7: "300 and 4000 nm" add wavelength afterwards, same as on p3l19 to make sure that wavelength is meant here

>>>>>>>>>>>>>>>>>>>>>>>>>>>>>>>>>>>>
Fixed: Corrected to clarify that these are wavelengths.
<<<<<<<<<<<<<<<<<<<<<<<<<<<<<<<<<<<<

P3l17: "… is required" Because it has not been done yet is not a convincing reason. What kind of consequences do you expect to derive from this study?

>>>>>>>>>>>>>>>>>>>>>>>>>>>>>>>>>>>>
Fixed: We have changed "required" to "useful", but we note that the other reviewer gave excellent reasons: "This study marks an important contribution for the remote sensing community, as different retrieval products from aircraft and satellites rely on an accurate knowledge of the multiangular reflectance of sea ice, which so far has been underrepresented in the literature. The retrieval products in need for a better representation of the surface reflectance anisotropy most notably include surface energy budget observations, but also include atmospheric retrievals that rely on surface reflectance corrections (e.g. cloud retrievals)."

<<<<<<<<<<<<<<<<<<<<<<<<<<<<<<<<<<<<<

Last paragraph of the introduction could be improved to reflect the outline in a better way.

>>>>>>>>>>>>>>>>>>>>>>>>>>>>>>>>>>>>>
Fixed: The results section has been re-ordered and matches the last paragraph now.
<<<<<<<<<<<<<<<<<<<<<<<<<<<<<<<<<<<<<

P3l32 / Fig. 1a: Do you really need Fig. 1a?

>>>>>>>>>>>>>>>>>>>>>>>>>>>>>>>>>>>>>
Figure 1a is a quick, clear, and unambiguous method to define the geometry and variables used in our paper. Whilst it is not absolutely essential it does add huge amounts of clarity to the work. Since no reason is given to remove it we have decided to keep it
<<<<<<<<<<<<<<<<<<<<<<<<<<<<<<<<<<<<<

Eq. (2) add "=pi*BRDF"

>>>>>>>>>>>>>>>>>>>>>>>>>>>>>>>>>>>>>
Fixed. This has been added, but it was written in the text one sentence below.
<<<<<<<<<<<<<<<<<<<<<<<<<<<<<<<<<<<<<

When I have understood correctly, the PlanarRad model was designed for aquatic radiative transfer. What justifies its use for calculating sea ice reflection properties? Is it a numerical model? Later (p5l9) Monte Carlo ray tracing is mentioned. Please explain a little more how this is related.

>>>>>>>>>>>>>>>>>>>>>>>>>>>>>>>>>>>>>
Nothing precludes the numerical model PlanarRad from being used for the study of sea ice. There are no theoretical barriers, it is freely available software for checking, and there is a good description of the model with references. The model is built for homogenous scattering and absorbing media. The owner of the model is an author on the paper which should give confidence. The model is designed in the same manner as the model Hydrolight which is well known in the field of hydrological optics. The ray tracing component of the model is for the consideration of surface roughness and again is explained in the methodology with references.
<<<<<<<<<<<<<<<<<<<<<<<<<<<<<<<<<<<<<

P4l10: "calculation of the BRDF" Actually, the model allows rather the simulation of the biconical reflection. So it's not a real BR(D)F which is shown in the following plots.

>>>>>>>>>>>>>>>>>>>>>>>>>>>>>>>>>>>>>
We agree with the reviewer that paper could be more precise in the definition of the properties evaluated, and we have added an initial clarification of this and modify the manuscript for consistency. The model used is designed in the

same way as the model HydroLight which is well-known in the field of hydrological optics. These models work with "quad-averaged radiances" where it assumed the radiance is constant over the solid angle subtended by each segment of the angular discretisation, both for "input" and "output". Therefore the 'BRDF's evaluated are biconical (geometry equivalent to case 5 in Table 2 of Scheapman-Strub et al. 2006), we have indicated this in the manuscript with extra text in methodology to make it very clear.

"Radiance is constant over the solid angle subtended by each segment of the angular discretisation, both for upwelling and downwelling and the directional reflectance is evaluated as conical-conical or biconical (geometry equivalent to case 5 in Table 2 of Schaepman-Strub et al. (2006)."

P2L19 - "Bidirectional Reflection Function (BRF) as an approximation of BRDF" was reworded to "Bidirectional Reflection Function (BRF) as an alternative to the BRDF".

References to "BRDF" and "BRF" have been corrected to "CCRF" or "conical-conical reflectance factor" or similar. The title of the manuscript has also been changed:-

"The effects of surface roughness on the calculated spectral (300 – 1400 nm) conical-conical reflectance factor (CCRF), as an alternative to bidirectional reflectance distribution function (BRDF), of bare sea ice "

And the following text has been added earlier on.

"Strictly the quantity, BRDF, cannot be measured and often other directional reflectance measurements are undertaken as an alternative or approximation of BRDF (Schaepman-Strub et al., 2006). There is a large number of terms in the literature for quantities that are measurable alternatives to BRDF that may not have been used uniformly and will be described herein as directional reflectance"
<<<<<<<<<<<<<<<<<<<<<<<<<<<<<<<<<<<<

P4l17+Fig. 1b and P4l28+Fig. 2: You could combine both figures as Fig 1a and Fig 1b. P5l3: Here the roughness parameter is without unit, but later and on p4l29 sigma is given in meters. Also in Fig. 3 its given without and with unit. Further, I suggest to move the definition of the roughness parameter to section 2.1.

>>>>>>>>>>>>>>>>>>>>>>>>>>>>>>>>>>>
Fixed: The roughness parameter should not have units. The following example is now given to help

"For example, if σ = 0.5, two points located 2mm apart would have their heights drawn from a normal distribution of mean zero and standard deviation of 1mm."

And with a helpful diagram to help visualise roughness along with:

"In the work presented here, 5 modelled surfaces were generated with an elevation standard deviation, σ = 0.001, 0.005, 0.01, 0.05 and 0.1 (Figure XX). The surfaces were generated using 10 rays per quad (4320 rays in total) with results averaged over 2000 surfaces. The roughness model being scale invariant, and the relative amplitude defined as 1 meter, the scale height of the roughness is 0.1, 0.5, 1, 5 and 10 cm."

Sigma is the elevation (height) standard deviation relative to the horizontal distance between two points on the surface and is unitless.
<<<<<<<<<<<<<<<<<<<<<<<<<<<<<<<<<<<<<

P5l11: "using 10 rays per quad" Could you explain, how is this number selected?

>>>>>>>>>>>>>>>>>>>>>>>>>>>>>>>>>>>>>
Fixed: It is a balance between reproducible simulation and computational effort. Further increasing the number of rays per quad did not increase the precision of the calculation beyond what is described in the paper.
<<<<<<<<<<<<<<<<<<<<<<<<<<<<<<<<<<<<<

P5l12: A scale height between 0.1 and 10 cm is chosen here. How do these numbers relate to real roughness features?

>>>>>>>>>>>>>>>>>>>>>>>>>>>>>>>>>>>>>
In the new section 2.3 the following text is introduced to explain that the roughness chosen were in keeping with the work of Tucker et al. (2013).

"To cover a wide range of conditions, a selection of five surface roughness parameters, defined by the standard deviation of the height of the surface were picked, with a standard deviation of 1 mm to 10 cm relative to two surface points 1m horizontally apart . The range of surface roughness is in agreement with observations reported in the literature for small scale roughnesses (e.g. Tucker et al., 2013)."
<<<<<<<<<<<<<<<<<<<<<<<<<<<<<<<<<<<<<

P5l14: The three sea ice types come somewhat out of the blue. Some more background should be already given in the introduction (see my first general comment).

>>>>>>>>>>>>>>>>>>>>>>>>>>>>>>>>>>>>>
Fixed: They were described in the methodology, in the section "Calculation of the CCRF of 3 types of sea ice with different roughness parameters", table 1 and the introduction. The section in the introduction now reads "In this work, the radiative-transfer model PlanarRad (Hedley, 2008, 2015) was used to model the CCRF (conical-conical reflectance factor) of three different types of sea ice from 300 to 1400 nm with varying thicknesses as a function of surface roughness in two steps. Firstly, the BRDF of three different types of sea ice with a thickness large enough to be optically thick was modelled with an increasing surface roughness. Secondly, the calculations performed in the

first step were repeated, but the optically thick thicknesses were replaced with fixed thicknesses of 50 cm and 100 cm for each type of sea ice. The optical properties of the three types of bare sea ice are chosen to represent multiyear sea ice, first year sea ice and melting sea ice and will be described in detail in the methodology."

<<<<<<<<<<<<<<<<<<<<<<<<<<<<<<<<<<<

P5l19: "mass-ratio of 1 ng/g" Did you see any effect of black carbon for this low number? From Marks and King (2014), Figure 3, I don't expect any significant contribution.

>>>>>>>>>>>>>>>>>>>>>>>>>>>>>>>>>>>
Fixed: A black carbon mass ratio of 1ng/g for this work and the previous work mentioned(Marks and King, 2014) was the smallest value of black carbon studied for these studies. There was not a smaller value to compare to. A small background mass ratio of black carbon is usual for radiative transfer modelling of snow and ice, and the work of Steve Warren (University of Washington) have shown it to be essential in all terrestrial snow and ices.
<<<<<<<<<<<<<<<<<<<<<<<<<<<<<<<<<<<

P5l25: "increases by 29.5%...up to 630.7%" I would give only integer numbers (here and elsewhere)

>>>>>>>>>>>>>>>>>>>>>>>>>>>>>>>>>>>
Fixed: The precision of such numbers has been reduced to integers.
<<<<<<<<<<<<<<<<<<<<<<<<<<<<<<<<<<<

P7l8: "...calculate the energy budget of the sea ice..." The sentence tries to motivate the direction of investigation. However, you should make this point already in the introduction, where the relation between satellite-based observations of the directional reflection, BRDF, BRF, ice albedo, and energy budget should be given.

>>>>>>>>>>>>>>>>>>>>>>>>>>>>>>>>>>>
Fixed: Referee 1 requested this argument be removed and it has.
<<<<<<<<<<<<<<<<<<<<<<<<<<<<<<<<<<<

P7l32: "moves lower on the hemisphere" – maybe better say that it is shifted to higher viewing zenith angles

>>>>>>>>>>>>>>>>>>>>>>>>>>>>>>>>>>>
Fixed: The phrase was be reworded to "...and moves to larger viewing zenith angle..."
<<<<<<<<<<<<<<<<<<<<<<<<<<<<<<<<<<<

P9l27: "to a larger zenith angle" – better write "viewing zenith angle" here

>>>>>>>>>>>>>>>>>>>>>>>>>>>>>>>>>>>
Fixed: The phrase was reworded as "...to a larger zenith angle..."
<<<<<<<<<<<<<<<<<<<<<<<<<<<<<<<<<<<

P11l1: The comparison with findings for snow surfaces should be extended. Give also quantitative results.

>>>>>>>>>>>>>>>>>>>>>>>>>>>>>>>>
Fixed: Now reads "Although not bare sea ice, there is some benefit in the comparison with the effect of surface roughness on the BRF of snow. Manninen et al. (2021) modelled the BRF of snow and found surface roughness of snow increased the back scattering at large solar zenith angle. Carlsen et al. (2020) measured the HDRF of Antarctic snow surfaces in the wavelength band 490–585 nm using a 180 degree fish-eye camera in an airborne platform whilst retrieving the surface roughness using an airborne laser scanner and found that the backscatter is enhanced over rougher surfaces concluding that shadows and changing the effective angle of incidence were responsible. Accepting that snow and sea ice are different materials with some similar optical similarities the findings presented here are consistent with the works of Manninen et al (2021) and Carlsen et al. (2020).
<<<<<<<<<<<<<<<<<<<<<<<<<<<<<<<<

P11 Section 4.2: I would shift this subsection to the model setup section. Statements related to the surface in general should be part of the introduction.

>>>>>>>>>>>>>>>>>>>>>>>>>>>>>>>>>
Fixed: This text was moved to the methodology.
<<<<<<<<<<<<<<<<<<<<<<<<<<<<<<<<<

Technical Corrections
P2l8: "rwith" – with

>>>>>>>>>>>>>>>>>>>>>>>>>>>>>>>>>
Fixed.
<<<<<<<<<<<<<<<<<<<<<<<<<<<<<<<<<<

P2l12 and other: insert a space before references "roughness(e.g. Manninen, 1997)" – roughness (e.g. Manninen, 1997)

>>>>>>>>>>>>>>>>>>>>>>>>>>>>>>>>>
Fixed.
<<<<<<<<<<<<<<<<<<<<<<<<<<<<<<<<<

P2l22: "snow kernals" – kernels

>>>>>>>>>>>>>>>>>>>>>>>>>>>>>>>>>
Fixed.
<<<<<<<<<<<<<<<<<<<<<<<<<<<<<<<<<

P2l31: "isotopic" – isotropic; Do you mean a Lambertian assumption here?

>>>>>>>>>>>>>>>>>>>>>>>>>>>>>>>>>
Fixed.
<<<<<<<<<<<<<<<<<<<<<<<<<<<<<<<<<

P3l1 and other: "dependance" – dependence

>>>>>>>>>>>>>>>>>>>>>>>>>>>>>>>>>
Fixed.
<<<<<<<<<<<<<<<<<<<<<<<<<<<<<<<<<

P4l4: "idea" – ideal

>>>>>>>>>>>>>>>>>>>>>>>>>>>>>>>>>
Fixed.
<<<<<<<<<<<<<<<<<<<<<<<<<<<<<<<<<

Figure 3: Please increase the font size of the axis labeling.

>>>>>>>>>>>>>>>>>>>>>>>>>>>>>>>>>
Fixed.
<<<<<<<<<<<<<<<<<<<<<<<<<<<<<<<<<

P6l3: "thickness" – ice thickness

>>>>>>>>>>>>>>>>>>>>>>>>>>>>>>>>>
Fixed.
<<<<<<<<<<<<<<<<<<<<<<<<<<<<<<<<<

Figure 4: Please increase the font size of the axis labeling.

>>>>>>>>>>>>>>>>>>>>>>>>>>>>>>>>>
Fixed.
<<<<<<<<<<<<<<<<<<<<<<<<<<<<<<<<<

P6l24: "... due to the large absorption in the ice dominating the signal ..." Sounds strange.

>>>>>>>>>>>>>>>>>>>>>>>>>>>>>>>>>
Fixed. Reworded to "For the three types of sea ice, the nadir value of the CCRF is strongly wavelength dependent due to the value of the absorption coefficient of ice increasing rapidly with wavelength and starting to change the interplay between scattering and absorption beyond 700 nm, and significantly lowering the CCRF"
<<<<<<<<<<<<<<<<<<<<<<<<<<<<<<<<<

5: please check color bar labeling

>>>>>>>>>>>>>>>>>>>>>>>>>>>>>>>>>
Fixed. Labelling corrected
<<<<<<<<<<<<<<<<<<<<<<<<<<<<<<<<<

7: roughness (sigma) is in wrong order, also check color bar labeling

>>>>>>>>>>>>>>>>>>>>>>>>>>>>>>>>>
Fixed. Roughness fixed – thank you. Colour bar labelling corrected
<<<<<<<<<<<<<<<<<<<<<<<<<<<<<<<<<

8: roughness (sigma) is in wrong order, also check color bar labeling

>>>>>>>>>>>>>>>>>>>>>>>>>>>>>>>>>
Fixed. Roughness fixed – thank you. Colour bar labelling corrected
<<<<<<<<<<<<<<<<<<<<<<<<<<<<<<<<<

P7l31: "as shown in Figure 6" – wrong reference

>>>>>>>>>>>>>>>>>>>>>>>>>>>>>>>>>
Fixed
<<<<<<<<<<<<<<<<<<<<<<<<<<<<<<<<<

P8l6: "inter-dependant" – inter-dependent

>>>>>>>>>>>>>>>>>>>>>>>>>>>>>>>>>
Fixed.
<<<<<<<<<<<<<<<<<<<<<<<<<<<<<<<<<

P8l20: "inter-dependant" – inter-dependent

>>>>>>>>>>>>>>>>>>>>>>>>>>>>>>>>>
Fixed.
<<<<<<<<<<<<<<<<<<<<<<<<<<<<<<<<<

P10l12: Start a new paragraph with "Miao et al. (2020) …"

>>>>>>>>>>>>>>>>>>>>>>>>>>>>>>>>>
Fixed.
<<<<<<<<<<<<<<<<<<<<<<<<<<<<<<<<<

P10l27: "a airborne" – an airborne

>>>>>>>>>>>>>>>>>>>>>>>>>>>>>>>>>
Fixed.
<<<<<<<<<<<<<<<<<<<<<<<<<<<<<<<<<

---

## Author Response (AR2)

**Reply to comment on tc-2021-366 (Lamare et al.)**

**Legend**
Reviewer
Author

I want to thank the authors for carefully revising the manuscript, I think it reads much more clearly now. The study is an important contribution for the remote sensing community, and the revisions helped to present the results more clearly. A more stringent use of the terminology of the reflectance quantities presented here makes the results immediately more applicable. This is important, as currently investigations of the directional reflectance of bare sea ice are underrepresented in the literature. Please find my remaining comments below.

>>>>>>>>>>>>>>>>>>>>>>>>>>>>>>>>>>>>>>>>>>>>>>>>>>>>>>>>>>>>>>>>>>>>>>>>>>>>
Thank you and thank you for taking the time to improve the manuscript.
<<<<<<<<<<<<<<<<<<<<<<<<<<<<<<<<<<<<<<<<<<<<<<<<<<<<<<<<<<<<<<<<<<<<<<<<<<<<

Title: I suggest removing the wavelength range in brackets in order to get a less bulky title. It is sufficient to mention the wavelength range in the abstract and throughout the text. Similarly, removing the abbreviations CCRF and BRDF from the title would foster reading comprehension and I think the paper will benefit from a clearer title.

>>>>>>>>>>>>>>>>>>>>>>>>>>>>>>>>>>>>>>>>>>>>>>>>>>>>>>>>>>>>>>>>>>>>>>>>>>>>
Done.
<<<<<<<<<<<<<<<<<<<<<<<<<<<<<<<<<<<<<<<<<<<<<<<<<<<<<<<<<<<<<<<<<<<<<<<<<<<<

Introduction: The introduction reads better than in the last version. However, I still have some comments:
(1) To focus the motivation of the study on the surface albedo and thus energy budget (for monitoring from satellite to inform models) is of course a very valid and relevant point. However, I am missing the connection of the CCRF/BRDF to other remote sensing retrievals (e.g. aerosol/cloud properties). These also have potentially large effects on the energy budget and a better representation of the directional reflectance can increase their accuracy, especially in the polar regions with large solar zenith angles.

>>>>>>>>>>>>>>>>>>>>>>>>>>>>>>>>>>>>>>>>>>>>>>>>>>>>>>>>>>>>>>>>>>>>>>>>>>>>

The following text and reference has been added to this short paragraph –
"The change in angular distribution of radiance at the top of the atmosphere, relative to the surface is significant, but the changes in the top of the atmosphere angular distribution of radiance owing to changes in atmospheric conditions are small(e.g. Hudson et al., 2010).".
<<<<<<<<<<<<<<<<<<<<<<<<<<<<<<<<<<<<<<<<<<<<<<<<<<<<<<<<<<<<<<<<<<<<<<<<<<<<

(2) The introduction 'jumps' a lot between thickness, roughness, wavelength considerations as well as between observations and modelling. Reordering of some of the points will help to foster reading comprehension.

>>>>>>>>>>>>>>>>>>>>>>>>>>>>>>>>>>>>>>>>>>>>>>>>>>>>>>>>>>>>>>>>>>>>>>>>>>>
The text was changed, especially regarding roughness and is now clearer.
<<<<<<<<<<<<<<<<<<<<<<<<<<<<<<<<<<<<<<<<<<<<<<<<<<<<<<<<<<<<<<<<<<<<<<<<<<<

(3) The introduction needs some paragraphs to break up the text, e.g. after the general motivation, after the definition of the CCRF/BRDF, and then potentially observations/modelling.

>>>>>>>>>>>>>>>>>>>>>>>>>>>>>>>>>>>>>>>>>>>>>>>>>>>>>>>>>>>>>>>>>>>>>>>>>>>
Paragraphs added.
<<<<<<<<<<<<<<<<<<<<<<<<<<<<<<<<<<<<<<<<<<<<<<<<<<<<<<<<<<<<<<<<<<<<<<<<<<<

Methods: I recommend breaking up the Methods a bit more, e.g. dedicating a separate subsection to the definition of the roughness parameter (new 2.3, and 2.3 becomes 2.4). It is great that it is included here now already, however I think it is easier to find and refer to if it is a separate subsection.

>>>>>>>>>>>>>>>>>>>>>>>>>>>>>>>>>>>>>>>>>>>>>>>>>>>>>>>>>>>>>>>>>>>>>>>>>>>
Done.
<<<<<<<<<<<<<<<<<<<<<<<<<<<<<<<<<<<<<<<<<<<<<<<<<<<<<<<<<<<<<<<<<<<<<<<<<<<

P2L33: large number of terms

>>>>>>>>>>>>>>>>>>>>>>>>>>>>>>>>>>>>>>>>>>>>>>>>>>>>>>>>>>>>>>>>>>>>>>>>>>>
Fixed.
<<<<<<<<<<<<<<<<<<<<<<<<<<<<<<<<<<<<<<<<<<<<<<<<<<<<<<<<<<<<<<<<<<<<<<<<<<<

P2L35: sea ice

>>>>>>>>>>>>>>>>>>>>>>>>>>>>>>>>>>>>>>>>>>>>>>>>>>>>>>>>>>>>>>>>>>>>>>>>>>>
Fixed.
<<<<<<<<<<<<<<<<<<<<<<<<<<<<<<<<<<<<<<<<<<<<<<<<<<<<<<<<<<<<<<<<<<<<<<<<<<<

P2L36: please mention some more recent observational studies as well, e.g. Goyens et al. (2018), Becker et al. (2022). These are highly relevant to the paper.

>>>>>>>>>>>>>>>>>>>>>>>>>>>>>>>>>>>>>>>>>>>>>>>>>>>>>>>>>>>>>>>>>>>>>>>>>>>
Fixed – references added.
<<<<<<<<<<<<<<<<<<<<<<<<<<<<<<<<<<<<<<<<<<<<<<<<<<<<<<<<<<<<<<<<<<<<<<<<<<<

P2L42: please specify the type of parameters used

>>>>>>>>>>>>>>>>>>>>>>>>>>>>>>>>>>>>>>>>>>>>>>>>>>>>>>>>>>>>>>>>>>>>>>>>>>>
Fixed – "Howeve, the study was limited to 2 spectral bands at 580–680 nm and 725–1000 nm and a single type of multi-year sea ice with parameters (salinity profile and air volume) obtained from Weeks and Ackley (1994)".
<<<<<<<<<<<<<<<<<<<<<<<<<<<<<<<<<<<<<<<<<<<<<<<<<<<<<<<<<<<<<<<<<<<<<<<<<<<

P2L56: please correct me if I am wrong, but wasn't the study by Marks et al. (2015) focused on Antarctic snow rather than sea ice? Do you mean Marks and King (2013) instead?

>>>>>>>>>>>>>>>>>>>>>>>>>>>>>>>>>>>>>>>>>>>>>>>>>>>>>>>>>>>>>>>>>>>>>>>
Fixed – well spotted thank you.
<<<<<<<<<<<<<<<<<<<<<<<<<<<<<<<<<<<<<<<<<<<<<<<<<<<<<<<<<<<<<<<<<<<<<<<

P4L100: close brackets

>>>>>>>>>>>>>>>>>>>>>>>>>>>>>>>>>>>>>>>>>>>>>>>>>>>>>>>>>>>>>>>>>>>>>>>
Fixed.
<<<<<<<<<<<<<<<<<<<<<<<<<<<<<<<<<<<<<<<<<<<<<<<<<<<<<<<<<<<<<<<<<<<<<<<

P4L113: affecting

>>>>>>>>>>>>>>>>>>>>>>>>>>>>>>>>>>>>>>>>>>>>>>>>>>>>>>>>>>>>>>>>>>>>>>>
Fixed.
<<<<<<<<<<<<<<<<<<<<<<<<<<<<<<<<<<<<<<<<<<<<<<<<<<<<<<<<<<<<<<<<<<<<<<<

P4L125-129: This should go into the introduction to introduce typical roughness of sea ice. And in the Methods, only explain the choice of roughness parameters and reference the Tucker et al. (2013) study.

>>>>>>>>>>>>>>>>>>>>>>>>>>>>>>>>>>>>>>>>>>>>>>>>>>>>>>>>>>>>>>>>>>>>>>>
Done.
<<<<<<<<<<<<<<<<<<<<<<<<<<<<<<<<<<<<<<<<<<<<<<<<<<<<<<<<<<<<<<<<<<<<<<<

P5L134: please consistently name it PlanarRad

>>>>>>>>>>>>>>>>>>>>>>>>>>>>>>>>>>>>>>>>>>>>>>>>>>>>>>>>>>>>>>>>>>>>>>>
All instances fixed.
<<<<<<<<<<<<<<<<<<<<<<<<<<<<<<<<<<<<<<<<<<<<<<<<<<<<<<<<<<<<<<<<<<<<<<<

Figure A3: the axis annotation font size needs to be larger, it is very hard to read at the moment.

>>>>>>>>>>>>>>>>>>>>>>>>>>>>>>>>>>>>>>>>>>>>>>>>>>>>>>>>>>>>>>>>>>>>>>>
The font size has been increased.
<<<<<<<<<<<<<<<<<<<<<<<<<<<<<<<<<<<<<<<<<<<<<<<<<<<<<<<<<<<<<<<<<<<<<<<

P6L163: is defined

>>>>>>>>>>>>>>>>>>>>>>>>>>>>>>>>>>>>>>>>>>>>>>>>>>>>>>>>>>>>>>>>>>>>>>>
Fixed.
<<<<<<<<<<<<<<<<<<<<<<<<<<<<<<<<<<<<<<<<<<<<<<<<<<<<<<<<<<<<<<<<<<<<<<<

P6L167: overly cautious/ conservative

>>>>>>>>>>>>>>>>>>>>>>>>>>>>>>>>>>>>>>>>>>>>>>>>>>>>>>>>>>>>>>>>>>>>>>>

Fixed – "conservative" used.

<<<<<<<<<<<<<<<<<<<<<<<<<<<<<<<<<<<<<<<<<<<<<<<<<<<<<<<<<<<<<<<<<<<<<<<

Caption Fig. A4: owing to changes *in* roughness are hard to discern…

>>>>>>>>>>>>>>>>>>>>>>>>>>>>>>>>>>>>>>>>>>>>>>>>>>>>>>>>>>>>>>>>>>>>>>>

Fixed.

<<<<<<<<<<<<<<<<<<<<<<<<<<<<<<<<<<<<<<<<<<<<<<<<<<<<<<<<<<<<<<<<<<<<<<<

Legend Fig. A4: please add the used roughness parameter to each line entry in the legend, not just for some of them.

>>>>>>>>>>>>>>>>>>>>>>>>>>>>>>>>>>>>>>>>>>>>>>>>>>>>>>>>>>>>>>>>>>>>>>>

Fixed.

<<<<<<<<<<<<<<<<<<<<<<<<<<<<<<<<<<<<<<<<<<<<<<<<<<<<<<<<<<<<<<<<<<<<<<<

P6L180: Please specify in the text whether the % changes given here are based on the peaks in the spectrum or are integrated quantities.

>>>>>>>>>>>>>>>>>>>>>>>>>>>>>>>>>>>>>>>>>>>>>>>>>>>>>>>>>>>>>>>>>>>>>>>

Fixed - "The nadir CCRF, at a wavelength of 500 nm, decreases by 21% when going from ..".

<<<<<<<<<<<<<<<<<<<<<<<<<<<<<<<<<<<<<<<<<<<<<<<<<<<<<<<<<<<<<<<<<<<<<<<

P7L190: CCRF instead of BRF

>>>>>>>>>>>>>>>>>>>>>>>>>>>>>>>>>>>>>>>>>>>>>>>>>>>>>>>>>>>>>>>>>>>>>>>

Fixed.

<<<<<<<<<<<<<<<<<<<<<<<<<<<<<<<<<<<<<<<<<<<<<<<<<<<<<<<<<<<<<<<<<<<<<<<

P8L234: CCRF instead of BRF

>>>>>>>>>>>>>>>>>>>>>>>>>>>>>>>>>>>>>>>>>>>>>>>>>>>>>>>>>>>>>>>>>>>>>>>

Fixed.

<<<<<<<<<<<<<<<<<<<<<<<<<<<<<<<<<<<<<<<<<<<<<<<<<<<<<<<<<<<<<<<<<<<<<<<

P11L324: explain acronym HDRF in the text here, as it is mentioned for the first time

>>>>>>>>>>>>>>>>>>>>>>>>>>>>>>>>>>>>>>>>>>>>>>>>>>>>>>>>>>>>>>>>>>>>>>>

Fixed, now reads "…Hemispherical Directional Reflectance Factor (HRDF)…".

<<<<<<<<<<<<<<<<<<<<<<<<<<<<<<<<<<<<<<<<<<<<<<<<<<<<<<<<<<<<<<<<<<<<<<<

P11L354: effect on…also remove double 'other'

>>>>>>>>>>>>>>>>>>>>>>>>>>>>>>>>>>>>>>>>>>>>>>>>>>>>>>>>>>>>>>>>>>>>>>>

Fixed both.

<<<<<<<<<<<<<<<<<<<<<<<<<<<<<<<<<<<<<<<<<<<<<<<<<<<<<<<<<<<<<<<<<<<<<<<

Acknowledgements: please change to the full link to the dataset
https://doi.org/10.5281/zenodo.5733402 as the current display of the doi is not working
directly.

>>>>>>>>>>>>>>>>>>>>>>>>>>>>>>>>>>>>>>>>>>>>>>>>>>>>>>>>>>>>>>>>>>>>>>>>>>
Fixed.
<<<<<<<<<<<<<<<<<<<<<<<<<<<<<<<<<<<<<<<<<<<<<<<<<<<<<<<<<<<<<<<<<<<<<<<<<

References

Goyens, C., Marty, S., Leymarie, E.,Antoine, D., Babin, M., & Bélanger, S.(2018). High angular
resolutionmeasurements of the anisotropyof reflectance of sea ice and snow.Earth and
Space Science,5, 30–47.https://doi.org/10.1002/2017EA000332

Becker, S., Ehrlich, A., Jäkel, E., Carlsen, T., Schäfer, M., and Wendisch, M.: Airborne
measurements of directional reflectivity over the Arctic marginal sea ice zone, Atmos. Meas.
Tech., 15, 2939–2953, https://doi.org/10.5194/amt-15-2939-2022, 2022.